# Neoadjuvant camrelizumab and apatinib combined with chemotherapy versus chemotherapy alone for locally advanced gastric cancer: a multicenter randomized phase 2 trial

Prospective evidence regarding the combination of programmed cell death (PD)−1 and angiogenesis inhibitors in treating locally advanced gastric cancer (LAGC) is limited. In this multicenter, randomized, phase 2 trial (NCT04195828), patients with gastric adenocarcinoma (clinical T2-4N + M0) were randomly assigned (1:1) to receive neoadjuvant camrelizumab and apatinib combined with nab-paclitaxel plus S-1 (CA-SAP) or chemotherapy SAP alone (SAP) for 3 cycles. The primary endpoint was the major pathological response (MPR), defined as <10% residual tumor cells in resection specimens. Secondary endpoints included R0 resection rate, radiologic response, safety, overall survival, and progression-free survival. The modified intention-to-treat population was analyzed (CA-SAP [$n = 51$] versus SAP [$n = 53$]). The trial has met pre-specified endpoints. CA-SAP was associated with a significantly higher MPR rate (33.3%) than SAP (17.0%, $P = 0.044$). The CA-SAP group had a significantly higher objective response rate (66.0% versus 43.4%, $P = 0.017$) and R0 resection rate (94.1% versus 81.1%, $P = 0.042$) than the SAP group. Non-surgical grade 3-4 adverse events were observed in 17 patients (33.3%) in the CA-SAP group and 14 (26.4%) in the SAP group. Survival results were not reported due to immature data. Camrelizumab and apatinib combined with chemotherapy as a neoadjuvant regimen was tolerable and associated with favorable responses for LAGC.

Gastric cancer (GC) is the fifth most frequently diagnosed malignancy and the fourth leading cause of cancer death worldwide, with the highest incidence and mortality rates reported in Eastern Asia[1]. Surgical resection is the mainstay of treatment for resectable GC; however, over 30% of patients with locally advanced gastric cancer (LAGC) relapse even after complete resection and adjuvant therapies[2,3]. Neoadjuvant treatment was introduced and has been widely applied to

improve the survival profiles of LAGC patients in the past 20 years[4,5]. To date, the exploration of the most effective neoadjuvant regimens continues.

A programmed cell death protein 1 (PD-1) inhibitor, which suppresses the interaction between PD-1 and its ligands (programmed cell death protein–ligand 1 [PD-L1] or PD-L2), has demonstrated encouraging antitumor activity in advanced GC. Based on the results of phase 3

e-mail: pingli811002@163.com; hcmlr2002@163.com

trials, a combination of the PD-1 inhibitor and chemotherapy exhibited extended clinical benefits[6,7] in comparison with PD-1 inhibitor monotherapy[8,9]. Moreover, neoadjuvant administration of PD-1 inhibitors with or without chemotherapy has been explored in two small, nonrandomized trials, with pathological complete response (pCR) rates of 19.4%[10] and 3.3%[11], respectively. These results suggest that PD-1 inhibitors should be used in combination with other systemic agents to strengthen their effectiveness.

Apatinib, an oral receptor tyrosine kinase inhibitor that selectively targets vascular endothelial growth factor (VEGF) receptor 2, has shown clinically significant efficacy in advanced or metastatic GC[12]. Our earlier phase 2 study revealed that apatinib combined with chemotherapy was effective and tolerable as a neoadjuvant treatment for LAGC[13]. Moreover, apatinib plus camrelizumab (a high-affinity humanized IgG4 monoclonal antibody targeting PD-1) has shown promising benefits in various malignancies[14,15]. We therefore hypothesized that apatinib and camrelizumab combined with chemotherapy might be beneficial in patients with LAGC.

Currently, paclitaxel-based chemotherapy has proven efficacy in LAGC[16] and was recommended as the first-line treatment[17]. Nanoparticle albumin-bound (nab)-paclitaxel, a 130 nm particle formulation consisting of paclitaxel and albumin nanoparticles linked by a noncovalent bond, improves the efficacy and safety of paclitaxel[18]. In this trial, we prespecified the regimen with nab-paclitaxel plus S-1 (SAP) as a control for two reasons: one was that a higher major pathological regression (MPR) rate and a low incidence of thrombocytopenia with SAP than with oxaliplatin plus S-1 (SOX) were observed in clinical practice[19], and the other was that nab-paclitaxel exhibited synergistic effects on both angiogenesis inhibitors and PD-1 inhibitors[20,21].

Here we reported the results of Arise-FJ-G005, a phase 2, multicenter, randomized controlled trial, that investigate the efficacy and safety of camrelizumab and apatinib combined with nab-paclitaxel plus S-1 versus nab-paclitaxel plus S-1 alone as neoadjuvant treatment for LAGC.

## Results

### Patients
Between June 18, 2020, and March 31, 2022, 106 patients were enrolled and underwent randomization at 5 centers. After excluding 2 patients who withdrew their consent after random assignment, 51 and 53 patients were treated with CA-SAP and SAP, respectively. The modified intention-to-treat (mITT) population consisted of these 104 patients. Two patients in the CA-SAP group and 3 patients in the SAP group did not receive surgery, and the remaining 99 patients comprised the per-protocol population. The flow diagram is provided in Fig. 1.

The median age of all patients was 63 years (first quartile-third quartile [Q1–Q3]: 57–68 years); 77 of 104 (74.0%) were men. Most of the patients had diffuse-type tumors ($n = 81$, 77.9%) and had cT4N+ disease ($n = 91$, 87.5%). Baseline characteristics are detailed in Table 1.

### Neoadjuvant and adjuvant treatments
Overall, 47 of 51 patients (92.2%) in the CA-SAP group and 48 of 53 patients (90.6%) in the SAP group completed the planned 3 cycles of neoadjuvant treatment; 4 patients in the CA-SAP group and 4 patients in the SAP group completed 2 cycles, and 1 patient in the SAP group completed 1 cycle. Four patients in the CA-SAP group discontinued preoperative treatment, of whom 3 experienced intolerable adverse events (AEs) and 1 had PD; 5 patients in the SAP group discontinued preoperative treatment, of whom 2 experienced intolerable AEs, 2 had PD, and 1 refused to continue the treatment.

Of 98 patients who underwent gastrectomy, 44 of 49 patients (89.8%) in the CA-SAP group and 47 of 49 patients (95.9%) in the SAP group received adjuvant treatment. Reasons for not starting adjuvant treatment in the CA-SAP group were poor performance status ($n = 1$) and patient request ($n = 4$). The median time to adjuvant treatment

from surgery was 36 days (Q1–Q3: 30–43 days) in the CA-SAP group and 35 days (Q1–Q3: 28–42 days) in the SAP group ($P = 0.338$). At the last follow-up (August 31, 2022), 22 of 44 patients (50.0%) in the CA-SAP group completed all 5 cycles of adjuvant treatment, 11 (25.0%) were still on treatment, and 11 (25.0%) discontinued the treatment; 22 of 47 patients (46.8%) in the SAP group completed all 5 cycles of adjuvant treatment, 12 (25.5%) were still on treatment, and 13 (27.7%) discontinued the treatment. Reasons for not completing adjuvant treatment in the CA-SAP group were AEs ($n = 6$), PD ($n = 1$), and patient request ($n = 4$). In the SAP group, the reasons were AEs ($n = 5$), PD ($n = 1$), and patient request ($n = 7$).

### Surgery
Forty-nine of 51 patients (96.1%) in the CA-SAP group and 50 of 53 patients (94.3%) in the SAP group underwent surgery, including 98 gastrectomies and 1 exploratory laparoscopy (SAP group). The median time between the last cycle of neoadjuvant treatment and surgery was 15 days (Q1–Q3: 14–21 days) in the CA-SAP group and 14 days (Q1–Q3: 14–17 days) in the SAP group ($P = 0.100$). The surgical characteristics and pathological findings of the patients who underwent gastrectomy are shown in Table 2. Of note, one patient in the CA-SAP group underwent palliative proximal gastrectomy due to acute bleeding.

### Efficacy
The results for tumor response are shown in Table 3. In the mITT population, a significantly higher proportion of patients achieved MPR (Tumor regression grade [TRG] 1a/b) in the CA-SAP group ($n = 17$, 33.3%; 95% CI: 19.9%–46.7%) than in the SAP group ($n = 9$, 17.0%; 95% CI: 6.5%–27.4%; $P = 0.044$, FDR-adjusted $P = 0.080$; Fig. 2a). Eight of 51 patients (15.7%; 95% CI: 5.4%–26.0%) in the CA-SAP group and 3 of 53 patients (5.7%; 95% CI: 0.0%–12.1%) in the SAP group achieved pCR (TRG 1a; $P = 0.089$, FDR-adjusted $P = 0.118$). In the per-protocol population, the MPR rate was also higher with CA-SAP (34.7%; 95% CI: 20.9%–48.5%) than with SAP (18.0%; 95% CI: 7.0%–29.0%; $P = 0.048$, FDR-adjusted $P = 0.080$). The pCR rates were 16.3% (95% CI: 5.6%–27.1%) and 6.0% (95% CI: 0%–12.8%), respectively, in the CA-SAP and SAP groups ($P = 0.094$, FDR-adjusted $P = 0.118$).

One hundred and three patients had evaluable radiologic results (Table 3), and one patient treated with CA-SAP did not receive radiologic assessment after neoadjuvant treatment. In the mITT population, an objective response was achieved in 33 of 51 patients (66.0%; 95% CI: 52.4%–79.6%) in the CA-SAP group and 23 of 53 patients (43.4%; 95% CI: 29.6%–57.2%) in the SAP group ($P = 0.017$, FDR-adjusted $P = 0.080$; Fig. 2b). The disease control rate (DCR) rate was 96.1% in the CA-SAP group and 96.2% in the SAP group ($P = 0.677$). In a comparison between the pretreatment and posttreatment clinical staging, T downstaging occurred in 52.9% of patients ($n = 27$) with CA-SAP and 32.1% of patients ($n = 17$) with SAP ($P = 0.025$, FDR-adjusted $P = 0.080$). N downstaging occurred in similar proportions of patients in both groups (25.5% versus 17.0%; $P = 0.206$, FDR-adjusted $P = 0.229$).

R0 resection was achieved in 48 of 51 patients (94.1%; 95% CI: 87.4%–100%) in the CA-SAP group and 43 of 53 patients (81.1%; 95% CI: 70.2%–92.0%) in the SAP group ($P = 0.042$, FDR-adjusted $P = 0.080$; Table 3). In the per-protocol population, the R0 resection rate was also significantly higher in the CA-SAP group (98.0%; 95% CI: 93.9%–100%) than in the SAP group (86.0%; 95% CI: 76.0%–96.0%; $P = 0.032$, FDR-adjusted $P = 0.080$).

### Subgroup analysis
We prespecified a set of subgroup analyses for the primary endpoint according to baseline characteristics in the per-protocol population (Fig. 3). Patients with an Eastern Cooperative Oncology Group (ECOG) performance status of 0 had significantly higher MPR rates with CA-SAP than with SAP (45.2% versus 14.7%; $P = 0.007$; $P$ for interaction $n = 0.031$). For intestinal-type tumors, the MPR rates were

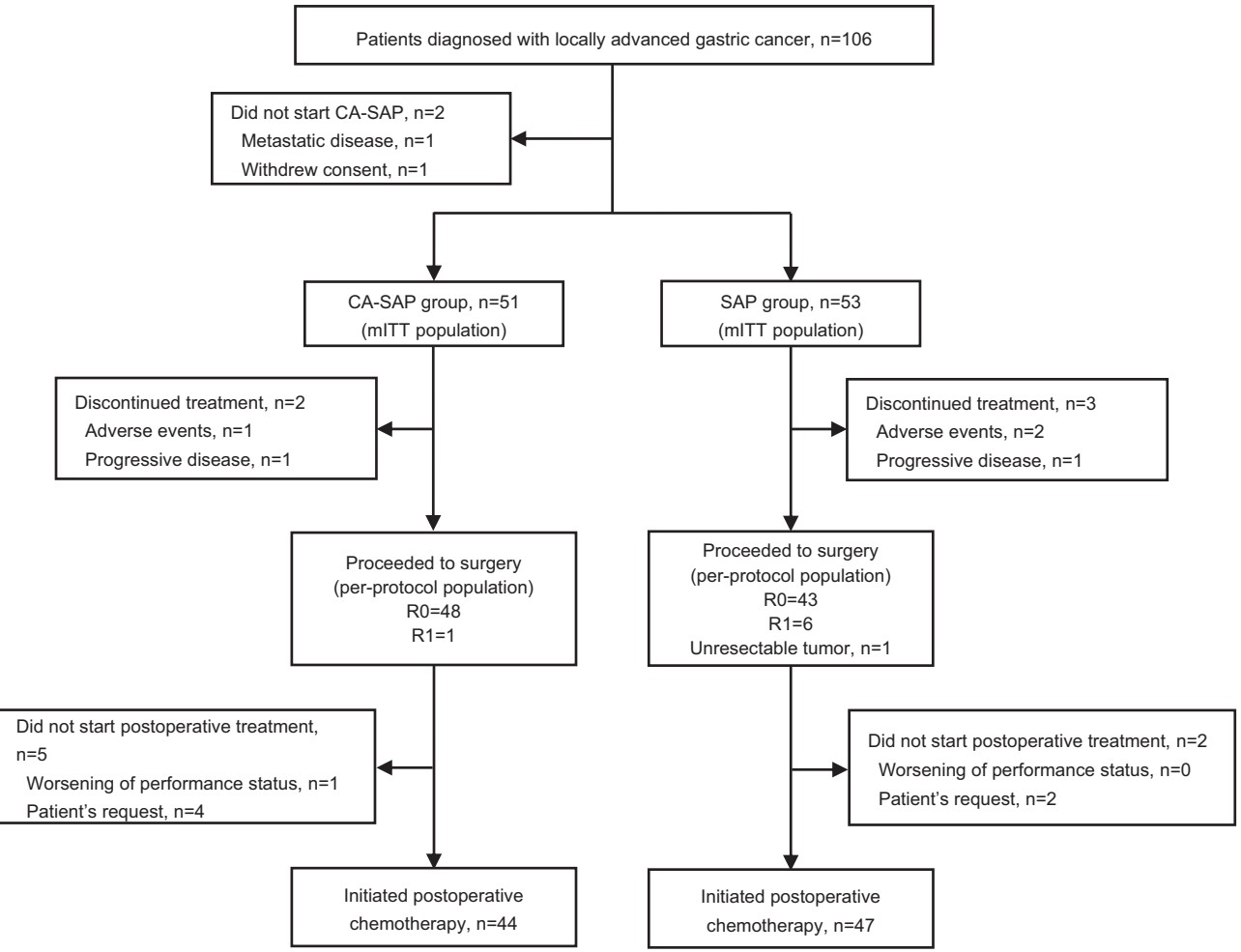

**Fig. 1 | Trial profile.** After excluding 2 patients who withdrew their consent after random assignment, 51 and 53 patients were treated with CA-SAP and SAP, respectively, and included in the mITT analysis. CA-SAP camrelizumab, apatinib, nab-paclitaxel, and S-1, SAP nab-paclitaxel and S-1, mITT modified intention-to-treat.

36.4% and 40.0% in the CA-SAP and SAP groups, respectively ($P = 0.608$). For diffuse-type tumors, the CA-SAP group showed a significantly higher MPR rate than the SAP group (34.2% vs. 12.5%, $P = 0.022$). However, this interaction did not reach statistical significance ($P$ for interaction = 0.227).

We explored the associations of MPR with PD-L1 expression and microsatellite instability (MSI) status. In the combined positive score (CPS) < 1% subgroup ($n = 45$), the MPR rates were 27.3% and 17.4% in the CA-SAP and SAP groups, respectively ($P = 0.331$). Among the 54 patients with a CPS ≥ 1%, the MPR rates were 40.7% and 18.5%, respectively, in the CA-SAP and SAP groups ($P = 0.068$). Among the 27 patients with a CPS ≥ 5%, the MPR rates were 50.0% and 27.3%, respectively, in the CA-SAP and SAP groups ($P = 0.107$; Supplementary Table 1). Compared with the SAP group, the CA-SAP group showed a trend toward a higher MPR rate in patients with MSI-H (66.7% [2 of 3 patients with CA-SAP] versus 0.0% [0 of 3 patients with SAP]) than in those with microsatellite stability (MSS; 32.6% [15 of 46 patients with CA-SAP] versus 19.1% [9 of 47 patients with SAP]).

### Safety
During the neoadjuvant treatment periods, the most common non-surgical AEs was leukopenia (CA-SAP: 72.5%; SAP: 71.7%) in both groups (Table 4). Seventeen of 51 patients (33.3%) in the CA-SAP group and 14 of 53 patients (26.4%) in the SAP group experienced at least one grade 3-4 AE ($P = 0.441$). The most common grade 3-4 AEs were leukopenia ($n = 7$, 13.7%), neutropenia ($n = 5$, 9.8%), and alanine

transaminase (ALT) elevation ($n = 5$, 9.8%) in the CA-SAP group and leukopenia ($n = 4$, 7.5%), neutropenia ($n = 3$, 5.7%), ALT elevation ($n = 3$, 5.7%), and aspartate aminotransferase (AST) elevation ($n = 3$, 5.7%) in the SAP group. Immune-related adverse events occurred in 10 patients (19.6%) in the CA-SAP group and in 1 patient (1.9%) in the SAP group, of which the most common event was hypothyroidism (Supplementary Table 2). All immune-related adverse events were grade 1 or 2.

Of 98 patients who underwent gastrectomy, postoperative recovery (all $P > 0.05$) and morbidity (20.4% [10 of 49 patients with CA-SAP] versus 12.2% [6 of 49 patients with SAP]; $P = 0.295$) were similar between the two groups (Supplementary Table 3). Most of the complications were categorized as Clavien–Dindo grade II. No reoperation or mortality within 30 days was observed.

### Discussion
The Arise-FJ-G005 study is a multicenter, randomized controlled trial evaluating the efficacy and safety of neoadjuvant anti–PD-1 immunotherapy and antiangiogenic therapy combined with chemotherapy versus chemotherapy alone in patients with LAGC. The study achieved the prespecified primary endpoint with a significantly higher MPR rate in the CA-SAP group (33.3%; 95% CI: 19.9%–46.7%) than in the SAP group (17.0%; 95% CI: 6.5%–27.4%). Analysis of secondary endpoints also revealed a significantly higher objective response rate (ORR) (66.0%) and R0 resection rate (94.1%) with an acceptable safety profile in patients with CA-SAP.

## Table 1 | Baseline characteristics of the modified intention-to-treat population

| Variable | CA-SAP group (n = 51) | SAP group (n = 53) |
|---|---|---|
| Age, years | 63 (57–68) | 63 (56–68) |
| Sex | | |
| Male | 42 (82.4) | 35 (66.0) |
| Female | 9 (17.6) | 18 (34.0) |
| ECOG performance status | | |
| 0 | 33 (64.7) | 36 (67.9) |
| 1 | 18 (35.3) | 17 (32.1) |
| Lauren classification | | |
| Intestinal | 11 (21.6) | 10 (18.9) |
| Diffuse | 39 (76.5) | 42 (79.2) |
| Unknown | 1 (2.0) | 1 (1.9) |
| Tumor location | | |
| Upper 1/3 | 22 (43.1) | 29 (54.7) |
| Middle 1/3 | 10 (19.6) | 6 (11.3) |
| Lower 1/3 | 11 (21.6) | 11 (20.8) |
| Mixed | 8 (15.7) | 7 (13.2) |
| Tumor size, mm | 65 (45–80) | 60 (50–75) |
| Borrmann type | | |
| II-III | 43 (84.3) | 48 (90.6) |
| IV | 8 (15.7) | 5 (9.4) |
| cT stage | | |
| T3 | 5 (9.8) | 8 (15.1) |
| T4 | 46 (90.2) | 45 (84.9) |
| PD-L1 expression (CPS) | | |
| <1 | 23 (45.1) | 23 (43.4) |
| ≥1 | 27 (52.9) | 28 (52.8) |
| Unknown | 1 (2.0) | 2 (3.8) |
| MSI status | | |
| MSS | 47 (92.2) | 48 (90.6) |
| MSI-High | 3 (5.9) | 3 (5.7) |
| Unknown | 1 (2.0) | 2 (3.8) |

Data are No. (%) or median (first quartile-third quartile [Q1–Q3]). Because of rounding, not all percentages add up to 100%.
CA-SAP camrelizumab, apatinib, nab-paclitaxel, and S-1, SAP nab-paclitaxel and S-1, ECOG Eastern Cooperative Oncology Group, PD-L1 programmed death-ligand 1, CPS combined positive score, MSI microsatellite instability, MSS microsatellite stable.

## Table 2 | Surgical and pathology findings

| Variable | CA-SAP group (n = 49) | SAP group (n = 49) |
|---|---|---|
| Surgical technology | | |
| Open | 1 (2.0) | 0 (0.0) |
| Laparoscopic | 45 (91.8) | 48 (98.0) |
| Robotic | 3 (6.1) | 1 (2.0) |
| Type of gastrectomy | | |
| Total | 39 (79.6) | 44 (89.8) |
| Distal | 9 (18.4) | 5 (10.2) |
| Proximal | 1 (2.0) | 0 (0.0) |
| Blood loss, mL | 35 (30–50) | 30 (30–50) |
| No. of lymph node metastasis | 1 (0–7) | 1 (0–7) |
| No. of lymph node harvested | 40 (29–55) | 40 (34–54) |
| Lymphovascular invasion | | |
| No | 25 (51.0) | 28 (57.1) |
| Yes | 24 (49.0) | 21 (42.9) |
| Neural invasion | | |
| No | 21 (42.9) | 19 (38.8) |
| Yes | 28 (57.1) | 30 (61.2) |
| ypT stage | | |
| T0 | 8 (16.3) | 3 (6.1) |
| T1 | 4 (8.2) | 5 (10.2) |
| T2 | 5 (10.2) | 5 (10.2) |
| T3 | 23 (46.9) | 23 (46.9) |
| T4a | 9 (18.4) | 13 (26.5) |
| ypN stage | | |
| N0 | 18 (36.7) | 21 (42.9) |
| N1 | 12 (24.5) | 8 (16.3) |
| N2 | 7 (14.3) | 7 (14.3) |
| N3 | 12 (24.5) | 13 (26.5) |
| ypM stage | | |
| M0 | 49 (100.0) | 47 (95.9) |
| M1 | 0 (0.0) | 2 (4.1) |
| Pathological response | | |
| TRG 1a | 8 (16.3) | 3 (6.1) |
| TRG 1b | 9 (18.4) | 6 (12.2) |
| TRG 2 | 10 (20.4) | 21 (42.9) |
| TRG 3 | 22 (44.9) | 19 (38.8) |

Data are No. (%) or median (first quartile-third quartile [Q1–Q3]). Because of rounding, not all percentages add up to 100%.
CA-SAP camrelizumab, apatinib, nab-paclitaxel, and S-1, SAP nab-paclitaxel and S-1, TRG tumor regression grade.

A neoadjuvant approach can downstage the tumor, improve the resectability, provide survival benefits[22], and has been widely used for the treatment of LAGC in Eastern and Western countries[23–25]. Pathological response is commonly used to evaluate the short-term effectiveness of neoadjuvant treatment[26]. Neoadjuvant FLOT has become a standard regimen in Europe due to the high pCR (16%) and MPR (37%) rates based on results from FLOT4-AIO[16]. However, differences in pharmacokinetics and tumor biology exist between Western and Asian populations[27], which may limit the application of FLOT in Asian populations. Although perioperative chemotherapy with SOX (RESOLVE trial) and DOS (PRODIGY trial) both improved progression-free survival (PFS) compared with adjuvant chemotherapy, the pCR rates of these two regimens (5.6% and 10.4%, respectively) were unsatisfactory[28,29]. Thus, there is an urgent need for a tolerable and more effective combination therapeutic regimen. Our results demonstrated that the CA-SAP group had higher MPR and pCR rates (33.3% and 17.0%, respectively) than the SAP group (17.0% and 5.7%, respectively). However, several nonrandomized trials of neoadjuvant immunochemotherapy have reported higher pCR and MPR rates than those in the CA-SAP group[10,30,31], while others have

reported lower rates[32,33]. To explore the additional effect of anti-angiogenesis therapy on neoadjuvant immunochemotherapy, we reviewed historical control patients receiving neoadjuvant camrelizumab plus SAP (C-SAP) during the same period (from 2020 to 2022) and met the eligibility criteria of this trial (Supplementary Table 4). The MPR (24.4%) and pCR rates (6.7%) of the C-SAP cohort was both lower than the CA-SAP group but higher than the SAP group (Supplementary Table 5). This finding suggested that the addition of apatinib to neoadjuvant immunochemotherapy might further improve the antitumor activity. CA-SAP also exhibited a higher pCR rate than apatinib plus SOX (6.3%) in our earlier study[13], indicating the synergistic antitumor activity of camrelizumab and apatinib. The immune suppressive nature of the tumor microenvironment is one of the most important reasons for primary resistance to immune checkpoint inhibitors and can be explained in part by the effects of

**Table 3 | Efficacy analysis in the modified intention-to-treat population**

| Variable | CA-SAP group (n = 51) | SAP group (n = 53) | P value | FDR-adjusted P value |
|---|---|---|---|---|
| Pathological response | | | | |
| TRG 0 (Complete) | 8 (15.7) | 3 (5.7) | | |
| TRG 1 (Subtotal) | 9 (17.6) | 6 (11.3) | | |
| TRG 2 (Partial) | 10 (19.6) | 21 (39.6) | | |
| TRG 3 (Minimal or none) | 22 (43.1) | 19 (35.8) | | |
| No gastrectomy | 2 (3.9) | 4 (7.5) | | |
| Major pathological response rate (%, 95% CI) | 33.3 (19.9–46.7) | 17.0 (6.5–27.4) | 0.044 | 0.080 |
| Complete response rate (%, 95% CI) | 15.7 (5.4–26.0) | 5.7 (0–12.1) | 0.089 | 0.118 |
| Radiologic response | | | | |
| CR | 3 (5.9) | 0 (0.0) | | |
| PR | 30 (58.8) | 23 (43.4) | | |
| SD | 16 (31.4) | 28 (52.8) | | |
| PD | 1 (2.0) | 2 (3.7) | | |
| Unidentified | 1 (2.0) | 0 (0.0) | | |
| Objective response rate (%, 95% CI) | 66.0 (52.4–79.6) | 43.4 (29.2–57.6) | 0.017 | 0.080 |
| Disease control rate (%, 95% CI) | 96.1 (90.6–100) | 96.2 (90.0–100) | 0.677 | 0.677 |
| Tumor downstaging | | | | |
| cT stage | Pre-treatment | Post-treatment | Pre-treatment | Post-treatment |
| T1 | 0 (0.0) | 1 (2.0) | 0 (0.0) | 0 (0.0) |
| T2 | 0 (0.0) | 10 (19.6) | 0 (0.0) | 5 (9.4) |
| T3 | 5 (9.8) | 17 (33.3) | 8 (15.1) | 19 (35.8) |
| T4 | 46 (90.2) | 22 (43.1) | 45 (84.9) | 29 (54.7) |
| Unidentified | 0 (0.0) | 1 (2.0) | 0 (0.0) | 0 (0.0) |
| T downstaging (%) | 27 (52.9) | 17 (32.1) | 0.025 | 0.080 |
| cN stage | Pre-treatment | Post-treatment | Pre-treatment | Post-treatment |
| N0 | 0 (0.0) | 13 (25.5) | 0 (0.0) | 9 (17.0) |
| N+ | 51 (100.0) | 36 (70.6) | 53 (100.0) | 44 (83.0) |
| Unidentified | 0 (0.0) | 1 (2.0) | 0 (0.0) | 0 (0.0) |
| N downstaging (%) | 13 (25.5) | 9 (17.0) | 0.206 | 0.229 |
| Surgical Fingdings | | | | |
| R0 resection rate (%, 95% CI) | 94.1 (87.4–100) | 81.1 (70.2–92.0) | 0.042 | 0.080 |

Data are No. (%). Because of rounding, not all percentages add up to 100%. P values were one-sided for efficacy analyses in Fisher's exact test and adjusted by controlling for the false discovery rate (FDR) using the Benjamini–Hochberg procedure.
*CA-SAP* camrelizumab, apatinib, nab-paclitaxel, and S-1, SAP nab-paclitaxel and S-1, *TRG* tumor regression grade, *CI* confidence interval, *CR* complete response, *PR* partial response, *SD* stable disease, *PD* progressive disease.

neoangiogenesis[34,35]. Anti-angiogenesis therapy can reverse this immune suppressive nature and has the potential to improve the therapeutic response to immunotherapy[36–38]. A two-by-two factorial randomized controlled trial should be conducted to further confirm this synergistic effect.

Paclitaxel-based chemotherapy has shown satisfactory efficacy and safety in the treatment of advanced gastric cancer[39–55] and shown non-inferior efficacy as compared with platinum-based chemotherapy in several randomized controlled trials[56–58]. A meta-analysis involving 1407 patients also supported the clinical efficacy of paclitaxel combined with S-1[59]. According to the Japanese gastric cancer treatment guidelines 2018 (5th edition), paclitaxel combined with S-1 or 5-FU, as well as platinum-based chemotherapy, were all considered as "Recommended regimens[17]". Our preliminary study also demonstrated a higher MPR rate with SAP than with SOX in clinical practice[19]. In addition, this trial aimed to explore the feasibility of immune checkpoint inhibitors (camrelizumab) and angiogenesis inhibitors (apatinib) in combination with chemotherapy as a neoadjuvant treatment for LAGC. Although neoadjuvant apatinib plus SOX has shown favorable efficacy in previous prospective studies, this regimen was associated with a high risk of thrombocytopenia[13,60,61]. This increased risk can be partly attributed to the use of oxaliplatin[62] and may lead to frequent treatment discontinuation[63]. Thus, this trial prespecified SAP as the chemotherapy regimen due to its low incidence of thrombocytopenia and high MPR rate.

Previous studies have demonstrated the predictive value of PD-L1 expression in response to anti–PD-1 immunotherapy in advanced GC. In the KEYNOTE-062 and CheckMate 649 trials, survival benefits in the addition of PD-1 inhibitors to chemotherapy were only demonstrated in patients with a higher CPS[6,9]. Our results also showed a trend toward a higher MPR rate in patients with a higher CPS (≥1% or ≥5%) than in those with a lower CPS in the CA-SAP group. Patients with CA-SAP who had a CPS of <1% showed an MPR rate (27.3%) similar to that reported with apatinib plus SOX (25.0%) in our earlier study[13]. These results suggest that adding PD-1 inhibitors to other antitumor agents might provide no benefit in patients with a lower CPS. Moreover, there is still no consensus regarding the association between PD-L1 expression and response to neoadjuvant chemotherapy[64–66]. Although the MPR rate was higher in the CPS ≥ 5% subgroup than in the CPS ≥ 1% and <1% subgroups in the SAP group, these differences did not reach statistical significance. Future studies are needed to confirm this relation. In addition, MSI status is a potential biomarker for GC treatment. In a retrospective study of 535 patients with LAGC, the MPR rate was significantly lower in patients with MSI-H (0%) than in those with MSS (16%)[67]. In the NEONIPIGA trial evaluating neoadjuvant immunotherapy in patients with MSI-H LAGC, 72.4% and 58.6% of patients, respectively, achieved MPR and pCR[68]. In our trial, only 6 patients had MSI-H tumors, with MPR rates of 66.7% and 0.0% in the CA-SAP and SAP groups, respectively. These results suggest the potential value of MSI status for selecting patients who may benefit more from anti–PD-1 immunotherapy; however, this prediction warrants further investigation due to the limited sample size.

The unique biological characteristics and tumor microenvironment of diffuse-type gastric cancer make it less sensitive to chemotherapy and immunotherapy[69,70]. The FLOT-4 trial demonstrated that patients with diffuse-type tumors exhibited lower pCR rates (both <3%) than those with intestinal-type tumors in both arms[16]. Likewise, among patients who were treated with neoadjuvant SAP, the MPR rate was significantly lower in diffuse-type tumors (12.5%) than in intestinal-type tumors (40.0%). In comparison, patients with diffuse-type tumors derived the highest benefit from neoadjuvant CA-SAP, with an MPR rate of 34.2%. A feasible explanation was that the introduction of apatinib altered the resistance profile of diffuse-type tumors. On one hand, anti-angiogenic therapy can improve the local hypoxia of diffuse-type tumors, thereby increasing sensitivity to chemotherapy and immunotherapy[71]. On the other hand, the immune-modulating properties of angiogenesis inhibitors may induce an immune-activated tumor microenvironment and enhance the efficacy of immunotherapy[72]. This finding might facilitate more individualized decision-making based on Lauren type. In addition, prespecified subgroup analysis showed a significant interaction between ECOG performance status and treatment regimen; patients with an ECOG performance status of 0 had a significantly higher MPR rate with CA-

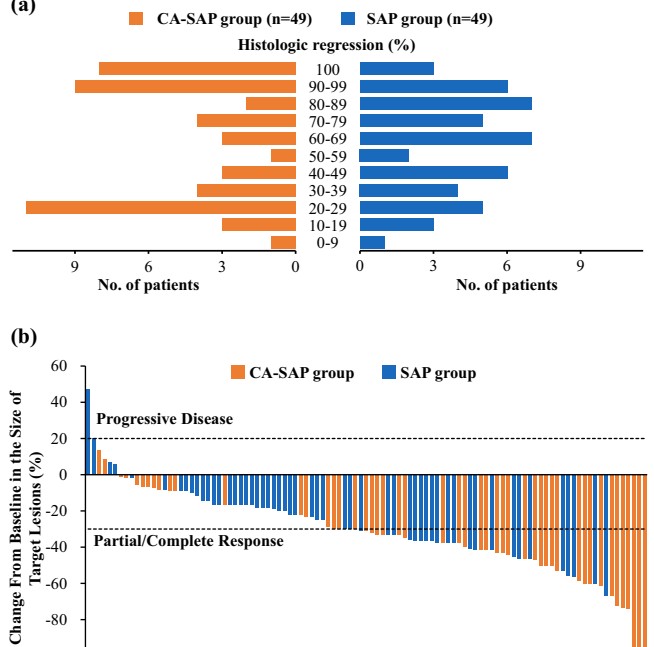

**Fig. 2 | Treatment response.** Results of pathological regression (**a**) and radiologic response assessed using the Response Evaluation Criteria in Solid Tumours (version 1.1) (**b**) in patients eligible for assessments (*n* = 98 and 103, respectively). CA-SAP camrelizumab, apatinib, nab-paclitaxel, and S-1, SAP nab-paclitaxel and S-1. Source data are provided as a Source data file.

SAP than those who had a performance status of 1 (45.2% versus 14.7%). In the KEYNOTE-059 and KEYNOTE-061 trials, better ECOG performance status was also associated with better overall survival with pembrolizumab[9,73]. Further investigation is needed to determine the potential predictive value of performance status on response to anti−PD-1 immunotherapy.

Although pathological response was the primary endpoint of this trial, the surrogacy of this pathological endpoint remains hotly debated[74,75]. In the FLOT4 trial, the superiority of FLOT in terms of pCR rates eventually translated into survival benefits[16]. The KEYNOTE 585 trial also demonstrated a statistically significant improvement in pCR rates in the chemotherapy plus pembrolizumab group; however, results did not meet statistical significance for event-free survival. Thus, active follow-up is needed to provide further insight into our findings. Nevertheless, pathological response could help to accelerate the process of testing new therapies as an early endpoint for predicting efficacy. Additionally, pathological response could be less susceptible to selection bias and less dependent on the quality of surgical resection compared with other endpoints. We therefore believe that pathological response could serve as an appropriate endpoint for neoadjuvant phase 2 trials.

Secondary efficacy endpoints included radiologic response and R0 resection rate. Because of the poor prognostic value of Response Evaluation Criteria in Solid Tumors (RECIST) response in patients with LAGC[76], both ORR and clinical downstaging were evaluated in this trial. We observed a higher ORR and a higher proportion of patients achieving T downstaging (66.0% and 52.9%, respectively) in the CA-SAP group than in the SAP group (43.4% and 32.1%, respectively). As previously reported, significant downstaging could provide favorable conditions for curative surgery[77]. In addition to the promising tumor response results, a remarkable improvement in the R0 resection rate was observed with CA-SAP. These results further support the favorable tumor response of neoadjuvant CA-SAP. Given the prognostic value of R0 resection and tumor downstaging[77–79], the advantages of CA-SAP in

these efficacy endpoints were expected to translate into improved survival outcomes.

Our results demonstrated a favorable safety profile of CA-SAP. The most common overall AE and grade 3-4 AE were both hematologic in patients with CA-SAP, which is in line with results reported with sintilimab plus CapeOx[10] and with apatinib plus SOX[13]. All AEs with potential immune etiology were categorized as grade 1–2 and were manageable according to the known safety management algorithm. No thromboembolism events were observed in the CA-SAP group, which was consistent with previous studies[12,80]. This finding showed a relatively low toxicity profile for apatinib, particularly in vascular toxicity. Chemotherapy may have direct or indirect effects on immune cells, leading to immune-related adverse reactions[81,82]. Similar to the KEYNOTE-061, KEYNOTE-062, and ATTRACTION-4 trials[7–9], one immune-related adverse reaction was also observed in the SAP group, but its incidence was obviously lower than the CA-SAP group.

Surgical outcomes were also manageable in both groups and comparable between them. Although an increased incidence of anastomotic leakage was observed in the CA-SAP group (8.2% versus 2.0%), this difference did not reach statistical significance (*P* = 0.201; Supplementary Table 3). In several prospective studies, apatinib plus neoadjuvant chemotherapy did not show a significant increase in the risk of anastomotic leakage[13,60,61]. Given the negative impact of VEGF inhibitors on anastomotic healing[83], we recommended stopping apatinib treatment at least 14 days before surgery and correcting hypoalbuminemia/anemia during the perioperative course.

Some limitations should be considered. First, our study was performed in an Asian population, and therefore, the effectiveness of CA-SAP should be validated in other populations. Second, although CA-SAP was demonstrated to be more effective than SAP, it remains unclear whether this superiority can translate into survival benefits. Active follow-up is needed to provide further insight into our findings. Third, the SAP chemotherapy regimen is not widely accepted as it was validated only in the Asian population, but we thought this regimen could be non-inferior to the standard regimens (e.g., oxaliplatin plus S-1). For example, the MPR and pCR rates (33.3% and 15.7%, respectively) in the CA-SAP group were similar to those reported by a recent nonrandomized trial investigating the neoadjuvant combination of camrelizumab, apatinib, and S-1 with or without oxaliplatin (26.3% and 15.8%, respectively)[84]. Nevertheless, a randomized controlled trial is needed to confirm the feasibility of camrelizumab and apatinib combined with platinum-based chemotherapy. Fourth, due to the two-arm design, it was unconvincing to demonstrate benefit of adding apatinib in the neoadjuvant treatment even with a historical control. Finally, the clinical response observed in this study should be further accompanied and explained with biomarkers and translational studies. These analyses are still ongoing in a post-hoc analysis. Nevertheless, we believe that our results can provide important information for further research and serve as preliminary data for a larger phase 3 trial.

In conclusion, camrelizumab and apatinib combined with nab-paclitaxel plus S-1 significantly increased the proportions of patients achieving pathological response, radiologic response, and R0 resection with acceptable safety compared with nab-paclitaxel plus S-1. This regimen might be a promising neoadjuvant treatment for patients with LAGC in the future, particularly in subpopulations with good performance status or diffuse-type tumors. An international, randomized phase 3 trial is needed to confirm our conclusions.

## Methods
### Trial design
We conducted a multicenter, randomized, open-label, phase 2 trial (Arise-FJ-G005) at 5 centers in China (Supplementary Fig. 1). The study protocol and all amendments were approved by the institutional review boards of the Fujian Medical University Union Hospital, Second Affiliated Hospital of Fujian Medical University, Zhongshan Hospital of

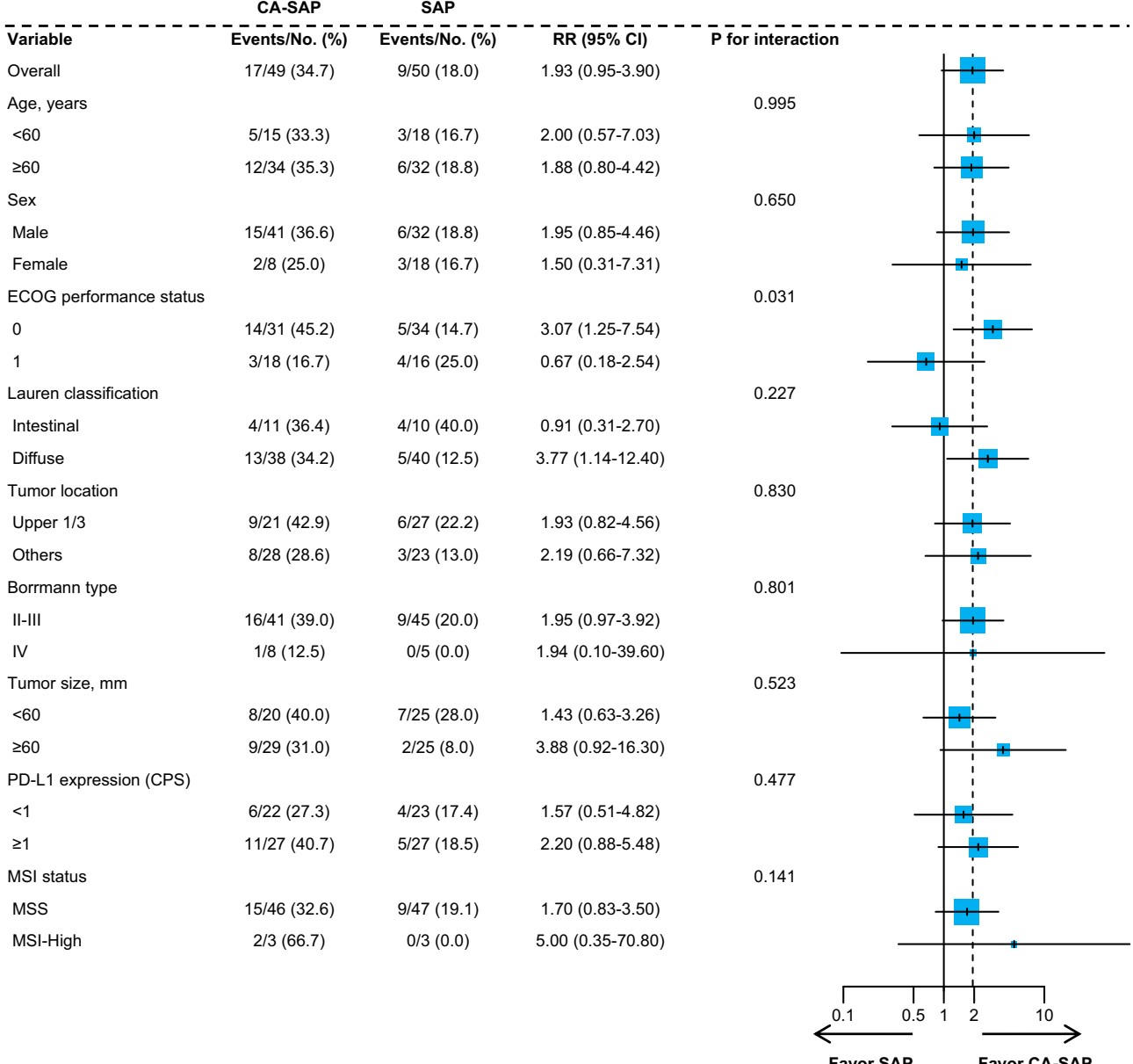

| Variable | CA-SAP Events/No. (%) | SAP Events/No. (%) | RR (95% CI) | P for interaction |
|---|---|---|---|---|
| Overall | 17/49 (34.7) | 9/50 (18.0) | 1.93 (0.95-3.90) | |
| Age, years | | | | 0.995 |
| <60 | 5/15 (33.3) | 3/18 (16.7) | 2.00 (0.57-7.03) | |
| ≥60 | 12/34 (35.3) | 6/32 (18.8) | 1.88 (0.80-4.42) | |
| Sex | | | | 0.650 |
| Male | 15/41 (36.6) | 6/32 (18.8) | 1.95 (0.85-4.46) | |
| Female | 2/8 (25.0) | 3/18 (16.7) | 1.50 (0.31-7.31) | |
| ECOG performance status | | | | 0.031 |
| 0 | 14/31 (45.2) | 5/34 (14.7) | 3.07 (1.25-7.54) | |
| 1 | 3/18 (16.7) | 4/16 (25.0) | 0.67 (0.18-2.54) | |
| Lauren classification | | | | 0.227 |
| Intestinal | 4/11 (36.4) | 4/10 (40.0) | 0.91 (0.31-2.70) | |
| Diffuse | 13/38 (34.2) | 5/40 (12.5) | 3.77 (1.14-12.40) | |
| Tumor location | | | | 0.830 |
| Upper 1/3 | 9/21 (42.9) | 6/27 (22.2) | 1.93 (0.82-4.56) | |
| Others | 8/28 (28.6) | 3/23 (13.0) | 2.19 (0.66-7.32) | |
| Borrmann type | | | | 0.801 |
| II-III | 16/41 (39.0) | 9/45 (20.0) | 1.95 (0.97-3.92) | |
| IV | 1/8 (12.5) | 0/5 (0.0) | 1.94 (0.10-39.60) | |
| Tumor size, mm | | | | 0.523 |
| <60 | 8/20 (40.0) | 7/25 (28.0) | 1.43 (0.63-3.26) | |
| ≥60 | 9/29 (31.0) | 2/25 (8.0) | 3.88 (0.92-16.30) | |
| PD-L1 expression (CPS) | | | | 0.477 |
| <1 | 6/22 (27.3) | 4/23 (17.4) | 1.57 (0.51-4.82) | |
| ≥1 | 11/27 (40.7) | 5/27 (18.5) | 2.20 (0.88-5.48) | |
| MSI status | | | | 0.141 |
| MSS | 15/46 (32.6) | 9/47 (19.1) | 1.70 (0.83-3.50) | |
| MSI-High | 2/3 (66.7) | 0/3 (0.0) | 5.00 (0.35-70.80) | |

0.1    0.5   1   2        10

← Favor SAP        Favor CA-SAP →

**Fig. 3 | Subgroup analysis of major pathological response in per-protocol population (CA-SAP [n = 49] vs. SAP [n = 50]).** Forest plots show the risk ratios (RRs) as centers, the upper and lower hinges represent the corresponding 95% confidence intervals (CIs). Interaction between agents was evaluated by likelihood ratio test, and P values were two sided at the 5% significance level. No adjustments were made for multiple comparisons. *CA-SAP* camrelizumab, apatinib, nab-paclitaxel, and S-1, *SAP* nab-paclitaxel and S-1, *ECOG* Eastern Cooperative Oncology Group, *PD-L1* programmed death-ligand 1, *CPS* combined positive score, *MSI* microsatellite instability, *MSS* microsatellite stable. Source data are provided as a Source Data file.

Xiamen University, Zhangzhou Municipal Hospital of Fujian Province, and The Affiliated Hospital of Putian University. All patients provided written informed consent. The study was performed in accordance with the Declaration of Helsinki and Good Clinical Practice guidelines. This study is registered with ClinicalTrials.gov, number NCT04195828. This study was reported in accordance with the Consolidated Standards of Reporting Trials (CONSORT) Guidelines. The original study protocol is available in the Supplementary Information as Supplementary Note 2.

**Participants**
Patients were eligible for enrollment if they were aged 18–75 years with at least one measurable lymph node with a short axis of ≥15 mm according to the RECIST (version 1.1)[85], histologically confirmed locally advanced gastric adenocarcinoma that was clinically T2 to T4 and M0

according to the 8th Edition of the American Joint Committee on Cancer (AJCC) Staging Manual, an ECOG performance status of 0 or 1, and adequate organ function. The main exclusion criteria were previous cancer therapy, history of malignancy within the past 5 years, or history of concurrent autoimmune disease. Complete inclusion and exclusion criteria are listed in Supplementary Table 6. The first patient was enrolled on June 18, 2020, and the last was recruited on March 31, 2022.

**Randomization and blinding**
A blinded statistician performed randomization with a list of randomly ordered treatment identifiers generated by SAS software, version 9.2 (SAS Institute). The randomized sequence was created for 1:1 allocation of 106 cases, 53 cases in each group, and was concealed from the investigators who screened and enrolled participants. The

**Table 4 | Summary of non-surgical adverse events**

| Adverse events | CA-SAP group (n = 51) | | SAP group (n = 53) | |
|---|---|---|---|---|
| | Grade 1–2 | Grade 3–4 | Grade 1–2 | Grade 3–4 |
| Hematologic | | | | |
| Neutropenia | 21 (41.2) | 5 (9.8) | 27 (50.9) | 3 (5.7) |
| Leukopenia | 30 (58.9) | 7 (13.7) | 35 (66.0) | 4 (7.5) |
| Thrombocytopenia | 7 (13.7) | 1 (2.0) | 0 (0.0) | 2 (3.8) |
| Anemia | 15 (29.4) | 1 (2.0) | 21 (39.6) | 0 (0.0) |
| Gastrointestinal | | | | |
| Nausea | 19 (37.3) | 0 (0.0) | 23 (43.4) | 0 (0.0) |
| Vomiting | 11 (21.6) | 0 (0.0) | 13 (24.5) | 0 (0.0) |
| Anorexia | 10 (19.6) | 0 (0.0) | 11 (20.8) | 1 (1.9) |
| Diarrhea | 14 (27.5) | 2 (3.9) | 8 (15.1) | 2 (3.8) |
| Bleeding | 8 (15.7) | 1 (2.0) | 8 (15.1) | 1 (1.9) |
| Liver | | | | |
| AST elevation | 11 (21.6) | 3 (5.9) | 8 (15.1) | 3 (5.7) |
| ALT elevation | 9 (17.6) | 5 (9.8) | 10 (18.9) | 3 (5.7) |
| Bilirubin increased | 8 (15.7) | 0 (0.0) | 8 (15.1) | 0 (0.0) |
| Cardio-renal | | | | |
| Hypertension | 10 (19.6) | 0 (0.0) | 2 (3.8) | 0 (0.0) |
| Proteinuria | 8 (15.7) | 0 (0.0) | 1 (1.9) | 0 (0.0) |
| Respiratory | | | | |
| Immune pneumonitis | 2 (3.9) | 0 (0.0) | 0 (0.0) | 0 (0.0) |
| Dermatologic | | | | |
| Rash | 9 (17.6) | 0 (0.0) | 5 (9.4) | 0 (0.0) |
| Hand–foot syndrome | 8 (15.7) | 0 (0.0) | 1 (1.9) | 0 (0.0) |
| Systemic | | | | |
| Fatigue | 15 (29.4) | 1 (2.0) | 19 (35.8) | 1 (1.9) |
| Fever | 10 (19.6) | 0 (0.0) | 5 (9.4) | 0 (0.0) |
| Others | | | | |
| Peripheral sensory neuropathy | 6 (11.8) | 0 (0.0) | 7 (13.2) | 1 (1.9) |
| Hypothyroidism | 7 (13.7) | 0 (0.0) | 1 (1.9) | 0 (0.0) |
| Stomatitis | 9 (17.6) | 1 (2.0) | 4 (7.5) | 0 (0.0) |

Data are No. (%).
*CA-SAP* camrelizumab, apatinib, nab-paclitaxel, and S-1, *SAP* nab-paclitaxel and S-1.

assignment was made by telephone contact or text messages after the patient met the eligibility criteria and signed the informed consent form. The study was open-label and no blinding was required. For randomization to be successfully implemented, the randomization sequence was concealed so that the investigators who screened and enrolled participants were not aware of the upcoming assignment. Patients and caregivers were not blinded to the treatment received. Outcome assessment for the primary endpoint was performed by two blinded pathologists. All statistical analyses were also performed by a blinded investigator.

## Treatments
Eligible patients were randomly assigned to receive camrelizumab (200 mg intravenously on day 1) and apatinib (250 mg orally once daily on days 1–21) combined with chemotherapy (nab-paclitaxel 125 mg/m$^2$ intravenously on days 2 and 9, S-1 40 to 60 mg orally twice daily depending on body surface area on days 1–14) or chemotherapy alone every 3 weeks for 3 preoperative cycles followed by 5 postoperative cycles. Dose modifications (e.g., dose interruption, delay, or reduction) were permitted in the presence of grade ≥3 hematologic or grade ≥2 nonhematologic AEs. The criteria for stopping treatment were

patient refusal, tumor progression, intolerable toxicity, or investigator's decision. An Independent Data Monitoring Committee (IDMC) monitored patient safety and study conduct.

After enrollment, tumor tissue samples were evaluated for PD-L1 expression and MSI status by a central laboratory in a blinded manner. PD-L1 expression was measured using the CPS, defined as the number of PD-L1–positive cells (tumor cells, lymphocytes, and macrophages) divided by the total number of tumor cells multiplied by 100, with the Ventana PD-L1 (SP263) immunohistochemistry assay. The MSI-high (MSI-H) status was defined as the loss of expression of at least one mismatch repair protein (MLH1, MSH1, MSH6, and PMS2). We performed MLH1 (ab92312, Abcam, 1: 250), MSH2 (ab52266, Abcam, 1: 250), MSH6 (ab92471, Abcam, 1: 250), PMS2 (ab110638, Abcam, 1: 250) immunohistochemical staining on the tissue.

Tumor assessments by means of contrast-enhanced computed tomography (CT) or magnetic resonance imaging (MRI) were performed after completion of the second cycle and before surgery. If tumor progression was demonstrated, surgery or other antitumor treatment could be administered at the investigator's discretion. Total or distal gastrectomy was scheduled 2 to 4 weeks after completion of the last cycle of neoadjuvant treatment. All surgical procedures, including the extent of gastric resection and D2 lymph node dissection, were performed according to the guidelines of the Japanese Research Society for the Study of Gastric Cancer[17]. All surgeons performed at least 200 gastrectomies for GC annually. Adjuvant treatment started 3 to 8 weeks after operation.

## Endpoints and assessments
The primary endpoint was the MPR rate, defined as the proportion of patients with <10% residual tumor cells in resection specimens[86]. Secondary endpoints included the pCR rate, R0 resection rate, radiologic response, safety, and survival.

Tumor regression grade was evaluated centrally using the Becker regression criteria, which are based on the percentage of vital tumor cells in the tumorous area and include the following categories: TRG 1a (no residual tumor cells), TRG 1b (<10% residual tumor cells), TRG 2 (10–50% residual tumor cells) and TRG 3 (>50% residual tumor cells)[86]. Radiologic response was evaluated using RECIST (version 1.1) by local radiologists, which is based on the short axis of the target lymph node(s) measured by CT or MRI findings and includes complete response (CR), partial response (PR), stable disease (SD), and progressive disease (PD)[85]. The ORR was defined as the proportion of patients with CR and PR, and the DCR was defined as the proportion of patients with CR, PR, and SD. R0 resection was defined as complete resection without macroscopic or microscopic residual disease, whereas R1 resection was defined as gross removal of tumors with microscopic resection margin involvement. Nonsurgical AEs were evaluated according to the Common Terminology Criteria for Adverse Events, version 5.0. Postoperative morbidity was evaluated according to the Clavien–Dindo classification[87]. Other secondary endpoints including overall survival and progression-free survival were not analyzed because the follow-up time was insufficient.

## Sample size and statistical analysis
Based on the assumption of MPR rates of 15% in the SAP group and 35% in the CA-SAP group, a sample size of 53 patients per group was required to detect improvement with 80% power and a type I error rate of 0.1 (Fisher's exact test), including a 5% dropout rate. The mITT population included all patients who were randomly assigned and received at least one dose of allocated treatment. The per-protocol population included patients in the mITT population who did not present major deviations from protocol. Efficacy analyses were performed in the mITT population and per-protocol population. Safety analyses were performed in all patients who received at least one dose of allocated treatment.

Continuous variables are presented as medians and interquartile ranges (Q1–Q3) and were compared using the Wilcoxon rank sum test. Categorical variables are presented as frequencies and percentages and were compared using the $\chi^2$ test or Fisher's exact test. Notably, the significance level was set to be 10% for efficacy analyses and 5% for other analyses. *P* values were one-sided for efficacy analyses in Fisher's exact test and were two-sided for other analyses. To address the issue of multiplicity, *P* values were adjusted by controlling for the false discovery rate (FDR) using the Benjamini–Hochberg procedure[88]. This post hoc adjustment was made for efficacy analyses, and no adjustment was made for other analyses which should be considered as explorative or descriptive. The study protocol prespecified a set of subgroup analysis according to baseline characteristics in the per-protocol population (Supplementary Note 2). Interaction between agents was evaluated by likelihood ratio test. Statistical significance of the interaction between baseline characteristics and treatment effect was assessed by comparing the logistic regression models with and without the interaction term. All statistical analyses were conducted with SPSS statistical software (version 21.0; SPSS Inc.) and R software (version 4.1.2; R Foundation for Statistical Computing).

### Reporting summary

Further information on research design is available in the Nature Portfolio Reporting Summary linked to this article.

## Data availability

The data supporting the findings in this study are available under controlled access due to data privacy laws related to patient consent for data sharing and the data should be used for research purposes only. All the original clinical data will be made available on request from the corresponding author (Huang CM) at any time in a de-identified manner. Request for data sharing will be handled in line with the data access and sharing policy of Fujian Medical University Union Hospital, which can be found in Supplementary Note 1. The original study protocol is available as Supplementary Note 2 in the Supplementary information file. The remaining data are available within the Article, Supplementary Information, or Source Data file. Source data are provided with this paper.

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

## Acknowledgements

We thank who have devoted a lot to this study, including nurses, pathologists, further-study doctors, and statisticians. This study was supported by the Construction Funds for "High-level Hospitals and Clinical Specialties" of Fujian Province (No. [2021]76, H.C.M.). The funding source had no role in the design and conduct of the study; collection, management, analysis, and interpretation of the data; preparation, review, or approval of the manuscript; and decision to submit the manuscript for publication.

## Author contributions

C.M.H. had full access to all the data in the study and takes responsibility for the integrity of the data and the accuracy of the data analysis. C.M.H. and J.X.L. obtained funding, conceived of and designed the study, and supervised the whole study. H.L.Z., K.Y., J.C.C., L.S.C., W.L., J.W.X., J.B.W., J.L., Q.Y.C., L.L.C., C.H.Z., and P.L. acquired the data, were responsible for study administration, and provided technical and material support. Y.H.T. interpreted and analyzed the data. Y.H.T. drafted the manuscript. C.M.H., J.X.L., C.H.Z., and P.L. critically revised the manuscript. All authors reviewed the manuscript and agreed to submit it for publication.

## Competing interests

The authors declare no competing interests.

## Additional information

Jian-Xian Lin[1,2,7], Yi-Hui Tang[1,7], Hua-Long Zheng[1,2,7], Kai Ye[3], Jian-Chun Cai[4], Li-Sheng Cai[5], Wei Lin[6], Jian-Wei Xie[1,2], Jia-Bin Wang[1,2], Jun Lu[1,2], Qi-Yue Chen[1,2], Long-Long Cao[1,2], Chao-Hui Zheng ®[1,2], Ping Li ®[1,2] ✉ & Chang-Ming Huang ®[1,2] ✉

[1]Department of Gastric Surgery, Fujian Medical University Union Hospital, Fuzhou, China. [2]Key Laboratory of Ministry of Education of Gastrointestinal Cancer, Fujian Medical University, Fuzhou, China. [3]Department of Gastrointestinal Surgery, Second Affiliated Hospital of Fujian Medical University, Quanzhou, China. [4]Department of Gastrointestinal Surgery, Zhongshan Hospital of Xiamen University, Xiamen, China. [5]Department of General Surgery, Zhangzhou Municipal Hospital of Fujian Province, Zhangzhou, China. [6]Department of Gastrointestinal Surgery, The Affiliated Hospital of Putian University, Putian, China. [7]These authors contributed equally: Jian-Xian Lin, Yi-Hui Tang, Hua-Long Zheng. ✉e-mail: pingli811002@163.com; hcmlr2002@163.com

