## [Peer Review File · Nature Communications]

Neoadjuvant Camrelizumab and Apatinib Combined with Chemotherapy versus Chemotherapy Alone for Locally Advanced Gastric Cancer: A Multicenter Randomized Phase 2 TrialREVIEWER COMMENTS

Reviewer #1 - Gastric cancer, immunotherapy (Remarks to the Author):

In this paper, the authors describe the ARISE trial, which is a multicentre phase II randomised trial conducted in China in operable gastric cancer. This trial examined a combination of camrelizumab and apatinib with chemotherapy including nab-paclitaxel+S1 versus chemotherapy alone before and after surgery in in locally advanced GC (T2-4 N+ M0). The primary end point of the trial was major pathological response.

As both PD-1 inhibition and anti-angiogenics are active in advanced disease, it makes sense (perhaps) to bring them into the operable setting. However, there are a few cautions. Why not use at least the standard of care in locally advanced cancer which is FLOT or platinum and 5FU or S1 (in Asia). Although MPR is increased – firstly we don't have a fair control arm to compare with, and second we don't know if it was apatinib or camrelizumab which improved the MPR. This is quite a major flaw in the study design.

Did lymph nodes meet criteria for RECIST? Primary tumour should not be measured.

Please comment in table on outcomes according to PD-L1

Please comment on rates of thromboembolism as there is a risk of this with apatinib.

Anastomotic breakdown is higher with antiVEGF. Please put this in context of other previous trials.

Please comment on recent data showing chemo + PD-1 did not improve OS despite improved pCR (KEYNOTE 585). Is pCR the best endpoint?

Reviewer #2 - Gastric cancer, immunotherapy (Remarks to the Author):

I am greatly honored to review this manuscript and the results of this study are of interest. But some issues still need to be addressed.

1. This study is registered on clinical trials.gov. The primary end point described on Clinical trials.gov was pCR. If pCR is used as the primary end point, the conclusions of this study should be modified as appropriate. If the protocol changes, the expected sample size should be adjusted accordingly. The upper age limit should be determined to be 70 or 75 years.

2. In the "Randomization and blinding" section, "The randomized sequence was created for 1:1 allocation of 106 cases, 53 cases in each group, and was concealed from the investigators who screened and enrolled participants. "The study is open-label, explaining the reason for the blinding.

3. The inclusion criteria are clinical stage T2-4, but there are no patients with T2 in the enrolled patients, and it is suggested to analyze the reasons.

4. To analyze the indications of proximal gastrectomy after neoadjuvant therapy.

In Table 2, what does 1 represent in the term lymph node metastasis? One lymph node metastasis? One patient had lymph node metastasis? Please explain.

5. In terms of safety, immune-related adverse reactions should be analyzed specifically.

6. In previous clinical trials of neoadjuvant anti-PD1 immunotherapy combined with chemotherapy, the PCR rate and MPR rate were higher than those in this study. Meanwhile, the addition of anti-angiogenesis drugs in this study did not significantly increase the PCR rate and MPR rate, and further analysis is recommended.

7. In the second paragraph of the discussion, "However, greater toxicities of this dose-intensive regimen, with 41% of patients experiencing treatment-related serious AEs, may limit its application in Asian populations. "This statement lacks evidence-based medical evidence. The author has previously mentioned that FLOT regimen neoadjuvant chemotherapy is recommended in China. DOI: 10.3760/cma.j.cn115610-20210622-00302.

8. The fifth paragraph of the discussion appears to be a restatement of the results and is suggested to be modified as appropriate.

Reviewer #3 - Biostatistics, clinical trials (Remarks to the Author):

This clinical trial was conducted appropriately, and the data analysis is mostly satisfactory.

Nonetheless, the following issues must be addressed for this report to be considered further.

1. The term, IQR (Interquartile range), is the difference of the lower and upper quartiles (thus, single number). Please use the correct term, "quartiles" when individual quartiles are listed.
2. Since this is a clinical trial, the baseline comparisons are not warranted. Please remove p-values for the baseline comparisons in Table 1 and (maybe) Table 2.
3. "alpha-level" (Line 130) is not a correct term. Use "Type I error rate".
4. Power analysis was conducted with one-sided type I error rate of 10%. Then why was the statistical significance threshold $P < 0.05$? These statements are contradictory. Please address.
5. Clear explanation is needed for how the interaction was tested. "Likelihood ratio test" implies comparison of two nested regression models, presumably one with and one without an interaction term.
6. Line 258: P for interaction is 0.227 means there is no statistical difference of treatment effects between intestinal and diffuse groups. Treatment difference is significant only in Diffuse group, but no statistically significant interaction indicates there is no evidence of a 'subgroup' difference. (Subgroup analysis using the interaction is the correct way.)
7. A large number of statistical tests were conducted without type I error rate adjustment. Some of the 'significant' effects have p-values barely below 0.05. Please address the issue of multiplicity and adjust the conclusions as needed.
8. Remove all 'useful' conclusions when p-values are large. For example, "comparable" line 243. The only conclusion one can draw from a large p-value is that the sample size was too small to detect a difference (effect). Also line 351 and perhaps other places.
9. Remove or edit eFigure 3. The data shown here are correlated (pre- and post- data on the same subjects), and a simple comparison of proportions is not appropriate. Analysis of these data may not be straightforward.

Reviewer #4 - Gastric cancer, immunotherapy (Remarks to the Author):

Suggestion: rejection

This study investigated the safety and efficacy of camrelizumab and apatinib(CA) combined with chemotherapy(nap-paclitaxel plus S-1, SAP) as a neoadjuvant regimen for LAGC.

Overall, this article is well organized and the regimen is novel, but there are several questions remained to be further explained. After reading the full manuscript, I have the following suggestions.

1. There remained major problems with the study design. Nowadays, for LAGC, more and more chemotherapy-immunotherapy combinations are entering clinical evaluation in the neoadjuvant setting, and several phase II trials have demonstrated the efficacy of adding PD-1 inhibitor. therefore, factorial analysis I think is necessarily considered in this trial design. C-SAP or A-SAP cohort is suggested to be added in this trial. At least, comparison with historical data should be added in the discussion part.

2. The SAP chemotherapy regimen is not the standard in clinical treatment of LAGC. In P3L86, the reasons to choose SAP in this study were not that convincing. The efficacy comparison of SOX and SAP should be cautiously explained since there were only small-sample size exploratory clinical trial results.

3. Diffuse-type gastric cancer (DGC) was considered to be insensitive to chemotherapy and immune checkpoint inhibitors, and MPR of DGC was lower in DGC than intestinal-type gastric cancer either in CA-SAP or SAP group in this study. But it's inspiring that DGC derive higher benefit from neoadjuvant CA-SAP than SAP, with an MPR rate of 34.2% versus 12.5%. Is there a potential molecular mechanism to explain this efficacy?

4. In SAP cohort, CPS \geq 5 subgroup showed a higher MPR rate with 27.3% than 18.5% and 17.4% in patients with CPS \geq 1 and CPS $<$ 1 subgroups, respectively. how do you explain the association between neoadjuvant chemotherapy efficacy and PD-L1 expression?

5. the use of anti-VEGF drugs in peri-operative setting may be associated with safety concerns. In this study, although the morbidity of anastomotic leakage was not statistically significant between two groups, CA-SAP group showed a trend toward more anastomotic

leakage (8.2% versus 2%). I think this should be considered in large randomized clinical trials in the future.

REVIEWER COMMENTS

Reviewer #1 - Gastric cancer, immunotherapy (Remarks to the Author):

In this paper, the authors describe the ARISE trial, which is a multicenter phase II randomized trial conducted in China in operable gastric cancer. This trial examined a combination of camrelizumab and apatinib with chemotherapy including nab-paclitaxel+S1 versus chemotherapy alone before and after surgery in in locally advanced GC (T2-4 N+ M0). The primary end point of the trial was major pathological response.

As both PD-1 inhibition and anti-angiogenics are active in advanced disease, it makes sense (perhaps) to bring them into the operable setting. However, there are a few cautions.

Why not use at least the standard of care in locally advanced cancer which is FLOT or platinum and 5FU or S1 (in Asia). Although MPR is increased – firstly we don't have a fair control arm to compare with, and second, we don't know if it was apatinib or camrelizumab which improved the MPR. This is quite a major flaw in the study design.

Response: Thank you for your valuable comment. Paclitaxel-based chemotherapy has shown satisfactory efficacy and safety in the treatment of advanced gastric cancer [1-17] and shown non-inferior efficacy as compared with platinum-based chemotherapy in several randomized controlled trials [18-20]. A meta-analysis involving 1407 patients also supported the clinical efficacy of paclitaxel combined with S-1 [21]. According to the Japanese gastric cancer treatment guidelines 2018 (5th edition), paclitaxel combined with S-1 or 5-FU, as well as platinum-based chemotherapy, were all considered as “Recommended regimens” [22]. Our preliminary study demonstrated a higher MPR rate with SAP than with SOX in clinical practice [23]. In addition, this trial aimed to explore the feasibility of immune checkpoint inhibitors

(camrelizumab) and angiogenesis inhibitors (apatinib) in combination with chemotherapy as a neoadjuvant treatment for LAGC. Although neoadjuvant apatinib plus SOX has shown favorable efficacy in previous prospective studies, this regimen was associated with a high risk of thrombocytopenia [24-26]. This increased risk can be partly attributed to the use of oxaliplatin [27] and may lead to frequent treatment discontinuation [28]. Thus, this trial prespecified SAP as the chemotherapy regimen due to its low incidence of thrombocytopenia [20] and high MPR rate. Nevertheless, it is a limitation of this trial to use paclitaxel-based chemotherapy instead of platinum-based chemotherapy as a control. Future studies are needed to investigate the feasibility of camrelizumab and apatinib combined with platinum-based chemotherapy. We have supplemented the above contents in the *Introduction* and *Discussion* parts of the revised manuscript.

Currently, camrelizumab plus apatinib has shown promising benefits in various malignancies as a novel combination strategy [29-31]. This might be explained by the synergistic antitumor activity of camrelizumab and apatinib. The immune suppressive nature of the tumor microenvironment is one of the most important reasons for primary resistance to immune checkpoint inhibitors and can be explained in part by the effects of neoangiogenesis [32-33]. Anti-angiogenesis therapy can reverse this immune suppressive nature and has the potential to improve the therapeutic response to immunotherapy [34-36]. We therefore hypothesized that apatinib and camrelizumab combined with chemotherapy might be beneficial in patients with LAGC. The results showed that the CA-SAP group had a significantly MPR rate than the SAP group. Furthermore, CA-SAP exhibited a higher pCR rate than both apatinib plus SOX (6.3%) [24] and camrelizumab plus FOLFOX (10%) in two nonrandomized studies

[37], thereby providing preliminary evidence for the hypothesis. A two-by-two factorial randomized controlled trial should be conducted to further confirm the synergistic antitumor activity of camrelizumab and apatinib. We have supplemented the above contents in the *Discussion* part of the revised manuscript.

Reference

- [1] Jeung HC, Rha SY, Kim YT, Noh SH, Roh JK, Chung HC. A phase II study of infusional 5-fluorouracil and low-dose leucovorin with docetaxel for advanced gastric cancer. *Oncology*. 2006;70(1):63-70.
- [2] Yoshida K, Ninomiya M, Takakura N, et al. Phase II study of docetaxel and S-1 combination therapy for advanced or recurrent gastric cancer. *Clin Cancer Res*. 2006;12(11 Pt 1):3402-3407.
- [3] Yamaguchi K, Shimamura T, Hyodo I, et al. Phase I/II study of docetaxel and S-1 in patients with advanced gastric cancer. *Br J Cancer*. 2006;94(12):1803-1808.
- [4] Im CK, Jeung HC, Rha SY, et al. A phase II study of paclitaxel combined with infusional 5-fluorouracil and low-dose leucovorin for advanced gastric cancer. *Cancer Chemother Pharmacol*. 2008;61(2):315-321.
- [5] Lee KS, Lee HY, Park EK, Jang JS, Lee SJ. A phase II study of leucovorin, 5-FU and docetaxel combination chemotherapy in patients with inoperable or postoperative relapsed gastric cancer. *Cancer Res Treat*. 2008;40(1):11-15.
- [6] Park SR, Kim HK, Kim CG, et al. Phase I/II study of S-1 combined with weekly docetaxel in patients with metastatic gastric carcinoma. *Br J Cancer*. 2008;98(8):1305-1311.
- [7] Lee JJ, Kim SY, Chung HC, et al. A multi-center phase II study of S-1 plus paclitaxel as first-line therapy for patients with advanced or recurrent unresectable gastric cancer. *Cancer Chemother Pharmacol*. 2009;63(6):1083-1090.
- [8] Zang DY, Yang DH, Lee HW, et al. Phase I/II trial with docetaxel and S-1 for patients with advanced or recurrent gastric cancer with consideration to age. *Cancer Chemother Pharmacol*. 2009;63(3):509-516.
- [9] Kakeji Y, Oki E, Egashira A, et al. Phase II study of biweekly docetaxel and S-1 combination therapy for advanced or recurrent gastric cancer. *Oncology*. 2009;77(1):49-52.
- [10] Kunisaki C, Takahashi M, Makino H, et al. Phase II study of biweekly docetaxel and S-1 combination chemotherapy as first-line treatment for advanced gastric cancer. *Cancer Chemother Pharmacol*. 2011;67(6):1363-1368.
- [11] Shigeyasu K, Kagawa S, Uno F, et al. Multicenter phase II study of S-1 and docetaxel combination chemotherapy for advanced or recurrent gastric cancer patients with peritoneal dissemination. *Cancer Chemother Pharmacol*. 2013;71(4):937-943.
- [12] Kosaka T, Akiyama H, Makino H, et al. Preoperative S-1 and docetaxel combination chemotherapy in patients with locally advanced gastric cancer. *Cancer Chemother Pharmacol*. 2014;73(2):281-285.
- [13] Koizumi W, Kim YH, Fujii M, et al. Addition of docetaxel to S-1 without platinum prolongs survival of patients with advanced gastric cancer: a randomized study (START). *J Cancer Res Clin Oncol*. 2014;140(2):319-328.
- [14] Oki E, Emi Y, Kusumoto T, et al. Phase II study of docetaxel and S-1 (DS) as neoadjuvant

- chemotherapy for clinical stage III resectable gastric cancer. *Ann Surg Oncol*. 2014;21(7):2340-2346.
- [15] Jiang H, Qian J, Zhao P, et al. A phase II study of biweekly S-1 and paclitaxel (SPA) as first-line chemotherapy in patients with metastatic or advanced gastric cancer. *Cancer Chemother Pharmacol*. 2015;76(1):197-203.
- [16] Kim YW, Kim MJ, Ryu KW, et al. A phase II study of perioperative S-1 combined with weekly docetaxel in patients with locally advanced gastric carcinoma: clinical outcomes and clinicopathological and pharmacogenetic predictors for survival. *Gastric Cancer*. 2016;19(2):586-596.
- [17] Kosaka T, Akiyama H, Miyamoto H, et al. Outcomes of preoperative S-1 and docetaxel combination chemotherapy in patients with locally advanced gastric cancer. *Cancer Chemother Pharmacol*. 2019;83(6):1047-1055.
- [18] Thuss-Patience PC, Kretzschmar A, Repp M, et al. Docetaxel and continuous-infusion fluorouracil versus epirubicin, cisplatin, and fluorouracil for advanced gastric adenocarcinoma: a randomized phase II study. *J Clin Oncol*. 2005;23(3):494-501.
- [19] Mochiki E, Ogata K, Ohno T, et al. Phase II multi-institutional prospective randomised trial comparing S-1+paclitaxel with S-1+cisplatin in patients with unresectable and/or recurrent advanced gastric cancer. *Br J Cancer*. 2012;107(1):31-36.
- [20] Dai YH, Yu XJ, Xu HT, et al. Nab-paclitaxel plus S-1 versus oxaliplatin plus S-1 as first-line treatment in advanced gastric cancer: results of a multicenter, randomized, phase III trial (GAPSO study). *Ther Adv Med Oncol*. 2022;14:17588359221118020.
- [21] Bian NN, Wang YH, Min GT. S-1 combined with paclitaxel may benefit advanced gastric cancer: Evidence from a systematic review and meta-analysis. *Int J Surg*. 2019;62:34-43.
- [22] Japanese Gastric Cancer Association. Japanese gastric cancer treatment guidelines 2018 (5th edition). *Gastric Cancer*. 2021;24(1):1-21.
- [23] Lin JL, Lin JX, Lin JP, et al. Safety and Efficacy of Camrelizumab in Combination With Nab-Paclitaxel Plus S-1 for the Treatment of Gastric Cancer With Serosal Invasion. *Front Immunol*. 2022;12:783243.
- [24] Lin JX, Xu YC, Lin W, et al. Effectiveness and Safety of Apatinib Plus Chemotherapy as Neoadjuvant Treatment for Locally Advanced Gastric Cancer: A Nonrandomized Controlled Trial. *JAMA Netw Open*. 2021;4(7):e2116240.
- [25] Zheng Y, Yang X, Yan C, et al. Effect of apatinib plus neoadjuvant chemotherapy followed by resection on pathologic response in patients with locally advanced gastric adenocarcinoma: A single-arm, open-label, phase II trial. *Eur J Cancer*. 2020;130:12-19.
- [26] Tang Z, Wang Y, Yu Y, et al. Neoadjuvant apatinib combined with oxaliplatin and capecitabine in patients with locally advanced adenocarcinoma of stomach or gastroesophageal junction: a single-arm, open-label, phase 2 trial. *BMC Med*. 2022;20(1):107.
- [27] Jardim DL, Rodrigues CA, Novis YAS, Rocha VG, Hoff PM. Oxaliplatin-related thrombocytopenia. *Ann Oncol*. 2012;23(8):1937-1942.
- [28] Al-Samkari H, Soff GA. Clinical challenges and promising therapies for chemotherapy-induced thrombocytopenia. *Expert Rev Hematol*. 2021;14(5):437-448.
- [29] Fan Y, Zhao J, Wang Q, et al. Camrelizumab Plus Apatinib in Extensive-Stage SCLC (PASSION): A Multicenter, Two-Stage, Phase 2 Trial. *J Thorac Oncol*. 2021;16(2):299-309.
- [30] Meng X, Wu T, Hong Y, et al. Camrelizumab plus apatinib as second-line treatment for advanced oesophageal squamous cell carcinoma (CAP 02): a single-arm, open-label, phase 2 trial. *Lancet Gastroenterol Hepatol*. 2022;7(3):245-253.

- [31] Liu J, Wang Y, Tian Z, et al. Multicenter phase II trial of Camrelizumab combined with Apatinib and Eribulin in heavily pretreated patients with advanced triple-negative breast cancer. *Nat Commun.* 2022;13(1):3011.
- [32] Binnewies M, Roberts EW, Kersten K, et al. Understanding the tumor immune microenvironment (TIME) for effective therapy. *Nat Med.* 2018;24(5):541-550.
- [33] Giraldo NA, Sanchez-Salas R, Peske JD, et al. The clinical role of the TME in solid cancer. *Br J Cancer.* 2019;120(1):45-53.
- [34] Voron T, Colussi O, Marcheteau E, et al. VEGF-A modulates expression of inhibitory checkpoints on CD8+ T cells in tumors. *J Exp Med.* 2015;212(2):139-148.
- [35] Gabrilovich DI, Chen HL, Girgis KR, et al. Production of vascular endothelial growth factor by human tumors inhibits the functional maturation of dendritic cells [published correction appears in *Nat Med* 1996 Nov;2(11):1267]. *Nat Med.* 1996;2(10):1096-1103.
- [36] Maenhout SK, Thielemans K, Aerts JL. Location, location, location: functional and phenotypic heterogeneity between tumor-infiltrating and non-infiltrating myeloid-derived suppressor cells. *Oncoimmunology.* 2014;3(10):e956579.
- [37] Liu Y, Han G, Li H, et al. Camrelizumab combined with FLOFOX as neoadjuvant therapy for resectable locally advanced gastric and gastroesophageal junction adenocarcinoma: Updated results of efficacy and safety. *J Clin Oncol.* 2021;39:4036-4036.

Did lymph nodes meet criteria for RECIST? Primary tumour should not be measured.

Response: Thank you for your valuable comment. In this trial, all enrolled patients had at least one measurable lymph node with a short axis of ≥ 15 mm according to the RECIST (version 1.1). Radiologic response was evaluated using RECIST (version 1.1) by local radiologists, which is based on the short axis of the target lymph node(s) measured by CT or MRI scans. We have supplemented the above contents in the *Methods* part of the revised manuscript.

Please comment in table on outcomes according to PD-L1

Response: Thank you for your valuable suggestion. Efficacy analysis according to PD-L1 expression (CPS) was detailed in **eTable 2**. In the CPS < 1 subgroup (n = 45), the MPR rates were 27.3% and 17.4% in the CA-SAP and SAP groups, respectively ($P = 0.331$). Among the 54 patients with a CPS ≥ 1 , the MPR rates were 40.7% and 18.5%, respectively, in the CA-

SAP and SAP groups ($P = 0.068$). Among the 27 patients with a CPS ≥ 5 , the MPR rates were 50.0% and 27.3%, respectively, in the CA-SAP and SAP groups ($P = 0.107$).

eTable 2. Efficacy analysis according to PD-L1 expression (CPS).

Variable	CA-SAP group	SAP group	P value
CPS <1%	n = 22	n = 23	
Major pathological response rate	27.3 (7.1-47.5)	17.4 (0.6-34.2)	0.331
Complete response rate	13.6 (0-29.2)	8.7 (0-21.2)	0.478
Objective response rate	68.2 (47.0-89.3)	43.5 (21.6-65.4)	0.086
R0 resection rate	95.5 (86.0-100)	87.0 (72.1-100)	0.321
CPS $\geq 1\%$	n = 27	n = 27	
Major pathological response rate	40.7 (20.9-60.5)	18.5 (2.9-34.2)	0.068
Complete response rate	18.5 (2.9-34.2)	3.7 (0-11.3)	0.096
Objective response rate	66.7 (47.7-85.7)	44.4 (24.4-64.5)	0.085
R0 resection rate	100 (NA)	85.2 (70.9-99.5)	0.056
CPS $\geq 5\%$	n = 16	n = 11	
Major pathological response rate	50.0 (22.5-77.5)	27.3 (0-58.7)	0.107
Complete response rate	18.8 (0-40.2)	9.1 (0-29.3)	0.455
Objective response rate	68.8 (43.2-94.3)	27.3 (0-58.7)	0.193
R0 resection rate	100 (NA)	81.8 (54.6-100)	0.157

NOTE: Data are percentages and 95% confidence intervals. CA-SAP = camrelizumab, apatinib, nap-paclitaxel, and S-1; SAP = nap-paclitaxel and S-1; NA = not applicable.

Please comment on rates of thromboembolism as there is a risk of this with apatinib.

Response: Thank you for your valuable suggestion. Thromboembolism is one of the most threatening adverse events directly related to antiangiogenic agents [1]. In this trial, no thromboembolism events were observed in the CA-SAP group, which was consistent with previous studies [2-3]. This finding showed a relatively low toxicity profile for apatinib, particularly in vascular toxicity. We have supplemented the above contents in the *Discussion* part of the revised manuscript.

Reference

[1] Riondino S, Del Monte G, Fratangeli F, Guadagni F, Roselli M, Ferroni P. Anti-Angiogenic Drugs, Vascular Toxicity and Thromboembolism in Solid Cancer. *Cardiovasc Hematol Agents Med Chem*.

2017;15(1):3-16.

[2] Li J, Qin S, Xu J, et al. Randomized, Double-Blind, Placebo-Controlled Phase III Trial of Apatinib in Patients With Chemotherapy-Refractory Advanced or Metastatic Adenocarcinoma of the Stomach or Gastroesophageal Junction. *J Clin Oncol*. 2016;34(13):1448-1454.

[3] Li J, Qin S, Wen L, et al. Safety and efficacy of apatinib in patients with advanced gastric or gastroesophageal junction adenocarcinoma after the failure of two or more lines of chemotherapy (AHEAD): a prospective, single-arm, multicenter, phase IV study. *BMC Med*. 2023;21(1):173.

Anastomotic breakdown is higher with antiVEGF. Please put this in context of other previous trials.

Response: Thank you for your valuable suggestion. The anti-angiogenic effect of VEGF inhibitors may negatively affect anastomotic healing in gastrointestinal surgery [1]. In a randomized controlled trial involving 1063 patients with esophagogastric cancer, an increased incidence of anastomotic leakage was observed in the chemotherapy plus bevacizumab group (24%) than in the chemotherapy alone group (10%) [2]. However, apatinib plus neoadjuvant chemotherapy did not show a significant increase in the risk of anastomotic leakage in several prospective studies [3-5]. In this trial, although an increased incidence of anastomotic leakage was observed in the CA-SAP group (8.2%), this difference did not reach statistical significance. Thus, we recommended stopping apatinib treatment at least 14 days before surgery and correcting hypoalbuminemia/anemia during the perioperative course, which could minimize the risk of anastomotic leakage in patients treated with apatinib. We have supplemented the above contents in the *Discussion* part of the revised manuscript.

Reference

[1] Nakamura H, Yokoyama Y, Uehara K, et al. The effects of bevacizumab on intestinal anastomotic healing in rabbits. *Surg Today*. 2016;46(12):1456-1463.

[2] Cunningham D, Stenning SP, Smyth EC, et al. Peri-operative chemotherapy with or without bevacizumab in operable oesophagogastric adenocarcinoma (UK Medical Research Council ST03): primary analysis results of a multicentre, open-label, randomised phase 2-3 trial. *Lancet Oncol*.

2017;18(3):357-370.

[3] Zheng Y, Yang X, Yan C, et al. Effect of apatinib plus neoadjuvant chemotherapy followed by resection on pathologic response in patients with locally advanced gastric adenocarcinoma: A single-arm, open-label, phase II trial. *Eur J Cancer*. 2020;130:12-19.

[1] Tang Z, Wang Y, Yu Y, et al. Neoadjuvant apatinib combined with oxaliplatin and capecitabine in patients with locally advanced adenocarcinoma of stomach or gastroesophageal junction: a single-arm, open-label, phase 2 trial. *BMC Med*. 2022;20(1):107.

[4] Lin JX, Xu YC, Lin W, et al. Effectiveness and Safety of Apatinib Plus Chemotherapy as Neoadjuvant Treatment for Locally Advanced Gastric Cancer: A Nonrandomized Controlled Trial. *JAMA Netw Open*. 2021;4(7):e2116240.

Please comment on recent data showing chemo + PD-1 did not improve OS despite improved pCR (KEYNOTE 585). Is pCR the best endpoint?

Response: Thank you for your valuable comment. Although pathological response was the primary endpoint of this trial, the surrogacy of this pathological endpoint remains hotly debated [1-2]. In the FLOT4 trial, the superiority of FLOT in terms of pCR rates eventually translated into survival benefits [3]. The KEYNOTE 585 trial also demonstrated a statistically significant improvement in pCR rates in the chemotherapy plus pembrolizumab group; however, results did not meet statistical significance for event-free survival. Thus, active follow-up is needed to provide further insight into our findings. Nevertheless, pathological response could help to accelerate the process of testing new therapies as an early endpoint for predicting efficacy. Additionally, pathological response could be less susceptible to selection bias and less dependent on the quality of surgical resection compared with other endpoints. We therefore believe that pathological response could serve as an appropriate endpoint for neoadjuvant phase 2 trials. We have supplemented the above contents in the *Discussion* part of the revised manuscript.

Reference

- [1] Conforti F, Pala L, Sala I, et al. Evaluation of pathological complete response as surrogate endpoint in neoadjuvant randomised clinical trials of early stage breast cancer: systematic review and meta-analysis. *BMJ*. 2021;375:e066381.
- [2] Huynh C, Sorin M, Rayes R, Fiset PO, Walsh LA, Spicer JD. Pathological complete response as a surrogate endpoint after neoadjuvant therapy for lung cancer. *Lancet Oncol*. 2021;22(8):1056-1058.
- [3] Al-Batran SE, Homann N, Pauligk C, et al. Perioperative chemotherapy with fluorouracil plus leucovorin, oxaliplatin, and docetaxel versus fluorouracil or capecitabine plus cisplatin and epirubicin for locally advanced, resectable gastric or gastro-oesophageal junction adenocarcinoma (FLOT4): a randomised, phase 2/3 trial. *Lancet*. 2019;393(10184):1948-1957.

Reviewer #2 - Gastric cancer, immunotherapy (Remarks to the Author):

I am greatly honored to review this manuscript and the results of this study are of interest. But some issues still need to be addressed.

1. This study is registered on clinical trials.gov. The primary end point described on Clinical trials.gov was pCR. If pCR is used as the primary end point, the conclusions of this study should be modified as appropriate. If the protocol changes, the expected sample size should be adjusted accordingly. The upper age limit should be determined to be 70 or 75 years.

Response: Thank you for your valuable comment. According to the latest version of the study protocol (version 1.3), we have made the following modifications on ClinicalTrials.gov: (1) The primary endpoint has been revised to MPR; 2) The upper age limit has been revised to 75 years.

2. In the "Randomization and blinding" section, "The randomized sequence was created for 1:1 allocation of 106 cases, 53 cases in each group, and was concealed from the investigators who screened and enrolled participants. "The study is open-label, explaining the reason for the blinding.

Response: Thank you for your valuable comment. The study was open-label and no blinding was required. For randomization to be successfully implemented, the randomization sequence was concealed so that the investigators who screened and enrolled participants were not aware of the upcoming assignment [1]. We have made corresponding modifications in the *Methods*

part of the revised manuscript.

Reference

[1] Viera AJ, Bangdiwala SI. Eliminating bias in randomized controlled trials: importance of allocation concealment and masking. *Fam Med.* 2007;39(2):132-137.

3. The inclusion criteria are clinical stage T2-4, but there are no patients with T2 in the enrolled patients, and it is suggested to analyze the reasons.

Response: Thank you for your valuable suggestion. This trial was designed to enroll patients with locally advanced gastric adenocarcinoma (clinical T2-4N+M0) who were suitable candidates for neoadjuvant treatment according to the guidelines [1]. However, patients who met eligibility criterion were all clinically staged as T3-4 during the screening process, and there were no patients with T2 disease in the enrolled patients. Nevertheless, we still described the *Method* part according to the original protocol.

Reference

[1] National Comprehensive Cancer, N. NCCN Clinical Practice Guidelines in Oncology (NCCN Guidelines®): Gastric Cancer. Version 2.2019. 2019.

4. To analyze the indications of proximal gastrectomy after neoadjuvant therapy.

Response: Thank you for your valuable suggestion. According to the Japanese gastric cancer treatment guidelines 2018 (5th edition), proximal gastrectomy can be considered for early proximal gastric cancer (cT1N0) where more than half of the distal stomach can be preserved [1]. For LAGC, there is currently no high-level evidence to confirm the feasibility of proximal gastrectomy. Thus, all patients with locally advanced proximal gastric cancer underwent total gastrectomy in this trial. Further studies are needed to explore the feasibility of proximal gastrectomy in patients with good response after neoadjuvant treatment. We have supplemented

the above contents in the *Methods* part of the revised manuscript.

Reference

[1] Japanese Gastric Cancer Association. Japanese gastric cancer treatment guidelines 2018 (5th edition). *Gastric Cancer*. 2021;24(1):1-21.

In Table 2, what does 1 represent in the term lymph node metastasis? One lymph node metastasis? One patient had lymph node metastasis? Please explain.

Response: In **Table 2**, 1 represents the median number of lymph node metastasis. We have revised “Lymph node metastasis” to “No. of lymph node metastasis”.

5. In terms of safety, immune-related adverse reactions should be analyzed specifically.

Response: Thank you for your valuable suggestion. Immune-related adverse events occurred in 10 patients (19.6%) in the CA-SAP group and in 1 patient (1.9%) in the SAP group, of which the most common event was hypothyroidism (**eTable 3**). All immune-related adverse events were grade 1 or 2, and no new safety signals emerged. We have supplemented the above contents in the *Results* part of the revised manuscript.

eTable 3. Treatment-related adverse events with potential immunological cause during neoadjuvant therapy.

Adverse Events	CA-SAP group (n=51)	SAP group (n=53)
	Any Grade	Any Grade
Hypothyroidism	7 (13.7)	1 (1.9)
Hyperthyroidism	0 (0.0)	0 (0.0)
Immune pneumonitis	2 (3.9)	0 (0.0)
Colitis	0 (0.0)	0 (0.0)
Hepatitis	2 (3.9)	0 (0.0)
Nephritis	0 (0.0)	0 (0.0)
Pancreatitis	0 (0.0)	0 (0.0)
Severe skin reactions	0 (0.0)	0 (0.0)
Adrenal insufficiency	0 (0.0)	0 (0.0)

Type 1 diabetes	0 (0.0)	0 (0.0)
Hypophysitis	1 (2.0)	0 (0.0)

NOTE. Data are No. (%).

6. In previous clinical trials of neoadjuvant anti-PD1 immunotherapy combined with chemotherapy, the PCR rate and MPR rate were higher than those in this study. Meanwhile, the addition of anti-angiogenesis drugs in this study did not significantly increase the PCR rate and MPR rate, and further analysis is recommended.

Response: Thank you for your valuable suggestion. As you mentioned, several trials of neoadjuvant immunochemotherapy reported higher pCR and MPR rates than those in this trial [1-3]. However, some trials also reported relatively lower pCR and MPR rates [4-5]. As a multicenter, randomized controlled trial, this study confirmed that CA-SAP was significantly associated with favorable responses as compared to SAP. CA-SAP also exhibited a higher pCR rate than apatinib plus SOX (6.3%) in our earlier study [6]. Future prospective evidence is still needed to confirm the feasibility of adding angiogenesis inhibitors to neoadjuvant immunochemotherapy. We have supplemented the above contents in the *Discussion* part of the revised manuscript.

Reference

- [1] Jiang H, Yu X, Li N, et al. Efficacy and safety of neoadjuvant sintilimab, oxaliplatin and capecitabine in patients with locally advanced, resectable gastric or gastroesophageal junction adenocarcinoma: early results of a phase 2 study. *J Immunother Cancer*. 2022;10(3):e003635.
- [2] Guo H, Ding P, Sun C, et al. Efficacy and safety of sintilimab plus XELOX as a neoadjuvant regimen in patients with locally advanced gastric cancer: A single-arm, open-label, phase II trial. *Front Oncol*. 2022;12:927781.
- [3] Yin Y, Lin Y, Yang M, et al. Neoadjuvant tislelizumab and tegafur/gimeracil/octeracil (S-1) plus oxaliplatin in patients with locally advanced gastric or gastroesophageal junction cancer: Early results of a phase 2, single-arm trial. *Front Oncol*. 2022;12:959295.
- [4] Liu Z, Liu N, Zhou Y, et al. Efficacy and safety of camrelizumab combined with FLOT versus FLOT alone as neoadjuvant therapy in patients with resectable locally advanced gastric and gastroesophageal

junction adenocarcinoma who received D2 radical gastrectomy: Data update. *J Clin Oncol.* 2022;40:16044-16044.

[5] Liu Y, Han G, Li H, et al. Camrelizumab combined with FLOFOX as neoadjuvant therapy for resectable locally advanced gastric and gastroesophageal junction adenocarcinoma: Updated results of efficacy and safety. *J Clin Oncol.* 2021;39:4036-4036.

[6] Lin JX, Xu YC, Lin W, et al. Effectiveness and Safety of Apatinib Plus Chemotherapy as Neoadjuvant Treatment for Locally Advanced Gastric Cancer: A Nonrandomized Controlled Trial. *JAMA Netw Open.* 2021;4(7):e2116240.

7. In the second paragraph of the discussion, “However, greater toxicities of this dose-intensive regimen, with 41% of patients experiencing treatment-related serious AEs, may limit its application in Asian populations. "This statement lacks evidence-based medical evidence. The author has previously mentioned that FLOT regimen neoadjuvant chemotherapy is recommended in China. DOI: 10.3760/cma.j.cn115610-20210622-00302.

Response: Thank you for your valuable comment. FLOT was recommended for neoadjuvant treatment by the guidelines of Chinese Society of Clinical Oncology (CSCO) [1]. However, differences in pharmacokinetics and tumor biology exist between Western and Asian populations [2]. Therefore, further studies are still needed to confirm the feasibility of FLOT as a neoadjuvant treatment in Asian populations. We have made corresponding modifications in the *Discussion* part of the revised manuscript.

Reference

[1] Wang FH, Zhang XT, Li YF, et al. The Chinese Society of Clinical Oncology (CSCO): Clinical guidelines for the diagnosis and treatment of gastric cancer, 2021. *Cancer Commun (Lond).* 2021;41(8):747-795.

[2] Huang RJ, Sharp N, Talamoa RO, Ji HP, Hwang JH, Palaniappan LP. One Size Does Not Fit All: Marked Heterogeneity in Incidence of and Survival from Gastric Cancer among Asian American Subgroups. *Cancer Epidemiol Biomarkers Prev.* 2020;29(5):903-909.

8. The fifth paragraph of the discussion appears to be a restatement of the results and is

suggested to be modified as appropriate.

Response: Thank you for your valuable suggestion. Secondary efficacy endpoints included radiologic response and R0 resection rate. Because of the poor prognostic value of RECIST response in patients with LAGC [1], both ORR and clinical downstaging were evaluated in this trial. We observed a higher ORR and a higher proportion of patients achieving T downstaging (66.0% and 52.9%, respectively) in the CA-SAP group than in the SAP group (43.4% and 32.1%, respectively). As previously reported, significant downstaging could provide favorable conditions for curative surgery [2]. In addition to the promising tumor response results, a remarkable improvement in the R0 resection rate was observed with CA-SAP. These results further support the favorable tumor response of neoadjuvant CA-SAP. Given the prognostic value of R0 resection and tumor downstaging [2-4], the advantages of CA-SAP in these secondary efficacy endpoints were expected to translate into improved survival outcomes. We have made corresponding modifications in the fifth paragraph of the *Discussion* part.

Reference

- [1] Kurokawa Y, Shibata T, Sasako M, et al: Validity of response assessment criteria in neoadjuvant chemotherapy for gastric cancer (JCOG0507-A). *Gastric Cancer* 17: 514-521, 2014
- [2] Prasad P, Sivaharan A, Navidi M, Fergie BH, Griffin SM, Phillips AW. Significance of neoadjuvant downstaging in gastric adenocarcinoma. *Surgery*. 2022;172(2):593-601.
- [3] Tu RH, Lin JX, Wang W, et al. Pathological features and survival analysis of gastric cancer patients with positive surgical margins: A large multicenter cohort study. *Eur J Surg Oncol*. 2019;45(12):2457-2464.
- [4] Levenson G, Voron T, Paye F, et al. Tumor downstaging after neoadjuvant chemotherapy determines survival after surgery for gastric adenocarcinoma. *Surgery*. 2021;170(6):1711-1717.

Reviewer #3 - Biostatistics, clinical trials (Remarks to the Author):

This clinical trial was conducted appropriately, and the data analysis is mostly satisfactory.

Nonetheless, the following issues must be addressed for this report to be considered further.

1. The term, IQR (Interquartile range), is the difference of the lower and upper quartiles (thus, single number). Please use the correct term, "quartiles" when individual quartiles are listed.

Response: Thank you for your valuable suggestion. We have revised “median (IQR)” to “median (first quartile-third quartile [Q1-Q3])” in the revised manuscript and tables.

2. Since this is a clinical trial, the baseline comparisons are not warranted. Please remove p-values for the baseline comparisons in Table 1 and (maybe) Table 2.

Response: Thank you for your valuable suggestion. We have removed p-values for the baseline comparisons in Tables 1-2.

3. "alpha-level" (Line 130) is not a correct term. Use "Type I error rate".

Response: Thank you for your valuable suggestion. We have revised “alpha-level” to “Type I error rate” in the *Methods* part of the revised manuscript.

4. Power analysis was conducted with one-sided type I error rate of 10%. Then why was the statistical significance threshold $P < 0.05$? These statements are contradictory. Please address.

Response: Thank you for your valuable comment. We have revised “A $P < 0.05$ was considered

statistically significant” to “The significance level was set to be 10% for efficacy analyses and 5% for other analyses” in the *Methods* part of the revised manuscript. We are very sorry for our negligence.

5. Clear explanation is needed for how the interaction was tested. "Likelihood ratio test" implies comparison of two nested regression models, presumably one with and one without an interaction term.

Response: Thank you for your valuable suggestion. Interaction between agents was evaluated by likelihood ratio test. Statistical significance of the interaction between baseline characteristics and treatment effect was assessed by comparing the logistic regression models with and without the interaction term. We have made corresponding modifications in the *Methods* part of the revised manuscript.

6. Line 258: P for interaction is 0.227 means there is no statistical difference of treatment effects between intestinal and diffuse groups. Treatment difference is significant only in Diffuse group, but no statistically significant interaction indicates there is no evidence of a 'subgroup' difference. (Subgroup analysis using the interaction is the correct way.)

Response: Thank you for your valuable comment. For intestinal-type tumors, the MPR rates were 36.4% and 40.0% in the CA-SAP and SAP groups, respectively ($P = 0.608$). For diffuse-type tumors, the CA-SAP group showed a significantly higher MPR rate than the SAP group (34.2% vs. 12.5%, $P = 0.022$). However, this interaction did not reach statistical significance (P for interaction = 0.227). We have made the above modifications in the *Results* part of the

revised manuscript.

7. A large number of statistical tests were conducted without type I error rate adjustment. Some of the 'significant' effects have p-values barely below 0.05. Please address the issue of multiplicity and adjust the conclusions as needed.

Response: Thank you for your valuable suggestion. To address the issue of multiplicity, P values were adjusted by controlling for the false discovery rate (FDR) for efficacy analyses using the Benjamini-Hochberg procedure [1]. No adjustment was made for the subgroup analysis which should be considered as explorative [2]. After adjustment, CA-SAP was significantly associated with a higher MPR rate, ORR, and T downstaging rate (FDR-adjusted $P < 0.1$). We have supplemented the above contents in the *Methods* part and made corresponding modifications in the *Abstract* and *Results* parts of the revised manuscript.

Table 3. Efficacy analysis in the modified intention-to-treat population.

Variable	CA-SAP group (n=51)	SAP group (n=53)	P value	FDR-adjusted P value
Pathological response				
TRG 0 (Complete)	8 (15.7)	3 (5.7)		
TRG 1 (Subtotal)	9 (17.6)	6 (11.3)		
TRG 2 (Partial)	10 (19.6)	21 (39.6)		
TRG 3 (Minimal or none)	22 (43.1)	19 (35.8)		
No gastrectomy	2 (3.9)	4 (7.5)		
Major pathological response rate (% , 95% CI)	33.3 (19.9-46.7)	17.0 (6.5-27.4)	0.044	0.080
Complete response rate (% , 95% CI)	15.7 (5.4-26.0)	5.7 (0-12.1)	0.089	0.118
Radiologic response				
CR	3 (5.9)	0 (0.0)		
PR	30 (58.8)	23 (43.4)		
SD	16 (31.4)	28 (52.8)		
PD	1 (2.0)	2 (3.7)		
Unidentified	1 (2.0)	0 (0.0)		
Objective response rate (% , 95% CI)	66.0 (52.4-79.6)	43.4 (29.2-57.6)	0.017	0.080

Disease control rate (% , 95% CI)	96.1 (90.6-100)	96.2 (90.0-100)	0.677	0.677
Tumor downstaging				
cT stage	Pre-treatment	Post-treatment	Pre-treatment	Post-treatment
T1	0 (0.0)	1 (2.0)	0 (0.0)	0 (0.0)
T2	0 (0.0)	10 (19.6)	0 (0.0)	5 (9.4)
T3	5 (9.8)	17 (33.3)	8 (15.1)	19 (35.8)
T4	46 (90.2)	22 (43.1)	45 (84.9)	59 (54.7)
Unidentified	0 (0.0)	1 (2.0)	0 (0.0)	0 (0.0)
T downstaging	27 (52.9)		17 (32.1)	0.025
				0.080
cN stage	Pre-treatment	Post-treatment	Pre-treatment	Post-treatment
N0	0 (0.0)	13 (25.5)	0 (0.0)	9 (17.0)
N+	51 (100.0)	36 (70.6)	53 (100.0)	44 (83.0)
Unidentified	0 (0.0)	1 (2.0)	0 (0.0)	0 (0.0)
N downstaging	13 (25.5)		9 (17.0)	0.206
				0.229
Surgical findings				
R0 resection rate (% , 95% CI)	94.1 (87.4–100)	81.1 (70.2–92.0)	0.042	0.080

NOTE: Data are No. (%). Because of rounding, not all percentages add up to 100%. CA-SAP = camrelizumab, apatinib, nap-paclitaxel, and S-1; SAP = nap-paclitaxel and S-1; FDR = false discovery rate; TRG = tumor regression grade; CI = confidence interval; CR = complete response; PR = partial response; SD = stable disease; PD = progressive disease.

Reference

- [1] Benjamini Y, Hochberg Y. Controlling the false discovery rate-a practical and powerful approach to multiple testing. *J Roy Stat Soc B Met.* 1995; 57: 289-300.
[2] Rothman KJ. No adjustments are needed for multiple comparisons. *Epidemiology.* 1990;1(1):43-46.

8. Remove all 'useful' conclusions when p-values are large. For example, "comparable" line 243. The only conclusion one can draw from a large p-value is that the sample size was too small to detect a difference (effect). Also line 351 and perhaps other places.

Response: Thank you for your valuable suggestion. According to your suggestion, we have made corresponding modifications in the revised manuscript. For example, we have revised “The DCR rates were comparable between the two groups (96.1% versus 96.2%; $P = 0.677$)” to “The DCR rate was 96.1% in the CA-SAP group and 96.2% in the SAP group ($P = 0.677$)”;

“Our results revealed that there were no significant differences between the CA-SAP and SAP groups in the nonsurgical AEs” has been revised to “Our results demonstrated a favorable safety profile of CA-SAP”.

9. Remove or edit eFigure 3. The data shown here are correlated (pre- and post- data on the same subjects), and a simple comparison of proportions is not appropriate. Analysis of these data may not be straightforward.

Response: According to your suggestion, we have removed **eFigure 3**.

Reviewer #4 - Gastric cancer, immunotherapy (Remarks to the Author):

Suggestion: rejection

This study investigated the safety and efficacy of camrelizumab and apatinib(CA) combined with chemotherapy(nap-paclitaxel plus S-1, SAP) as a neoadjuvant regimen for LAGC. Overall, this article is well organized and the regimen is novel, but there are several questions remained to be further explained. After reading the full manuscript, I have the following suggestions.

Response: Thank you for your valuable comments and suggestions. We have studied the comments carefully and have made modifications and corrections, which helped us to improve the quality of the manuscript. This trial demonstrated that neoadjuvant PD-1 and angiogenesis inhibitors combined with chemotherapy significantly increased the proportions of patients achieving pathological response, radiologic response, and R0 resection compared with chemotherapy alone. This regimen could be a promising neoadjuvant treatment for patients with LAGC. For aggressive diffuse-type tumors, this combination therapy provided more benefits than chemotherapy alone. This trial provides a new treatment option for LAGC, contributing high-level evidence to the clinical management of gastric cancer. Detailed revisions are as follows:

1. There remained major problems with the study design. Nowadays, for LAGC, more and more chemotherapy-immunotherapy combinations are entering clinical evaluation in the neoadjuvant setting, and several phase II trials have demonstrated the efficacy of adding PD-1 inhibitor.

therefore, factorial analysis I think is necessarily considered in this trial design. C-SAP or A-SAP cohort is suggested to be added in this trial. At least, comparison with historical data should be added in the discussion part.

Response: Thank you for your valuable suggestion. We reviewed historical control patients receiving camrelizumab combined with nab-paclitaxel plus S-1 (C-SAP) as a neoadjuvant treatment during the same period (from 2020 to 2022) and met the eligibility criteria of this trial. Patient characteristics of the C-SAP cohort were similar with the CA-SAP and SAP groups ($P > 0.05$, **eTable 5**). The MPR (24.4%) and pCR rates (6.7%) of the C-SAP cohort was both lower than the CA-SAP group but higher than the SAP group (**eTable 6**). This finding suggested that the addition of angiogenesis inhibitors to neoadjuvant immunochemotherapy might further improve antitumor activity, which indicated the synergistic effects of PD-1 and angiogenesis inhibitors. A two-by-two factorial randomized controlled trial should be conducted to further confirm this synergistic antitumor activity. We have supplemented the above contents in the *Discussion* part of the revised manuscript.

eTable 5. Clinicopathological Characteristics of the Historical Cohort.

Variable	C-SAP cohort	P value ^a	P value ^b
Age, years ^c	60 (54-68)	0.220	0.367
Sex ^c			
Male	30 (66.7)	0.077	0.948
Female	15 (33.3)		
ECOG performance status ^c			
0	28 (62.2)	0.801	0.555
1	17 (37.8)		
Lauren classification ^c			
Intestinal	14 (31.1)	0.407	0.253
Diffuse	29 (64.4)		
Unknown	2 (4.4)		
Tumor location ^c			

Upper 1/3	18 (40.0)		
Middle 1/3	10 (22.2)		
Lower 1/3	11 (24.4)	0.958	0.389
Mixed	6 (13.3)		
Tumor size, mm ^c	58 (45-70)	0.130	0.270
Borrmann type ^c			
II-III	41 (91.1)		
IV	4 (9.9)	0.315	0.926
cT stage ^c			
T3	3 (6.7)		
T4	42 (93.3)	0.579	0.188
Type of gastrectomy ^d			
Total	31 (77.5)		
Distal	9 (22.5)	0.811	0.113
Lymphovascular invasion ^d			
No	22 (55.0)		
Yes	18 (45.0)	0.708	0.839
Neural invasion ^d			
No	23 (57.5)		
Yes	17 (42.5)	0.169	0.078
ypT stage ^d			
T0	2 (5.0)		
T1	8 (20.0)		
T2	4 (10.0)	0.575	0.894
T3	17 (42.5)		
T4a	9 (22.5)		
ypN stage ^d			
N0	16 (40.0)		
N1	5 (12.5)		
N2	9 (22.5)	0.527	0.791
N3	10 (25.0)		
ypM stage ^d			
M0	40 (100.0)		
M1	0 (0.0)	1.000	0.500

^a Characteristics were compared between the C-SAP cohort and the CA-SAP group.

^b Characteristics were compared between the C-SAP cohort and the SAP group.

^c Characteristics were compared between the whole C-SAP cohort (n=45) and the modified intention-to-treat sets (CA-SAP: n=51; SAP: n=53).

^d Characteristics were compared between the C-SAP cohort proceeding to gastrectomy (n=40) and the per-protocol sets (CA-SAP: n=49; SAP: n=49).

NOTE: Data are No. (%) or median (first quartile-third quartile [Q1-Q3]). Because of rounding, not all percentages add up to 100%. C-SAP = camrelizumab, nap-paclitaxel, and S-1; CA-SAP = camrelizumab, apatinib, nap-paclitaxel, and S-1; SAP = nap-paclitaxel and S-1; ECOG = Eastern Cooperative Oncology Group.

eTable 6. Efficacy analysis of the Historical Cohort.

Variable	C-SAP cohort (n=45)	CA-SAP group (n=51)	SAP group (n=53)
Major pathological response rate	24.4 (11.4-37.5)	33.3 (19.9-46.7)	17.0 (6.5-27.4)
Complete response rate	6.7 (0-14.2)	15.7 (5.4-26.0)	5.7 (0-12.1)
Objective response rate	51.1 (35.9-66.3)	66.0 (52.4-79.6)	43.4 (29.2-57.6)
R0 resection rate	88.9 (79.3-98.4)	94.1 (87.4–100)	81.1 (70.2–92.0)

NOTE: Data are percentages and 95% confidence intervals. C-SAP = camrelizumab, nap-paclitaxel, and S-1; CA-SAP = camrelizumab, apatinib, nap-paclitaxel, and S-1; SAP = nap-paclitaxel and S-1.

2. The SAP chemotherapy regimen is not the standard in clinical treatment of LAGC. In P3L86, the reasons to choose SAP in this study were not that convincing. The efficacy comparison of SOX and SAP should be cautiously explained since there were only small-sample size exploratory clinical trial results.

Response: Thank you for your valuable comment. Paclitaxel-based chemotherapy has shown satisfactory efficacy and safety in the treatment of advanced gastric cancer [1-17] and shown non-inferior efficacy as compared with platinum-based chemotherapy in several randomized controlled trials [18-20]. A meta-analysis involving 1407 patients also supported the clinical efficacy of paclitaxel combined with S-1 [21]. According to the Japanese gastric cancer treatment guidelines 2018 (5th edition), paclitaxel combined with S-1 or 5-FU, as well as platinum-based chemotherapy, were all considered as “Recommended regimens” [22]. Our preliminary study demonstrated a higher MPR rate with SAP than with SOX in clinical practice [23]. In addition, this trial aimed to explore the feasibility of immune checkpoint inhibitors (camrelizumab) and angiogenesis inhibitors (apatinib) in combination with chemotherapy as a neoadjuvant treatment for LAGC. Although neoadjuvant apatinib plus SOX has shown

favorable efficacy in previous prospective studies, this regimen was associated with a high risk of thrombocytopenia [24-26]. This increased risk can be partly attributed to the use of oxaliplatin [27] and may lead to frequent treatment discontinuation [28]. Thus, this trial prespecified SAP as the chemotherapy regimen due to its low incidence of thrombocytopenia [20] and high MPR rate. Nevertheless, it is a limitation of this trial to use paclitaxel-based chemotherapy instead of platinum-based chemotherapy as a control. Future studies are needed to investigate the feasibility of camrelizumab and apatinib combined with platinum-based chemotherapy. We have supplemented the above contents in the *Introduction* and *Discussion* parts of the revised manuscript.

Reference

- [2] Jeung HC, Rha SY, Kim YT, Noh SH, Roh JK, Chung HC. A phase II study of infusional 5-fluorouracil and low-dose leucovorin with docetaxel for advanced gastric cancer. *Oncology*. 2006;70(1):63-70.
- [3] Yoshida K, Ninomiya M, Takakura N, et al. Phase II study of docetaxel and S-1 combination therapy for advanced or recurrent gastric cancer. *Clin Cancer Res*. 2006;12(11 Pt 1):3402-3407.
- [4] Yamaguchi K, Shimamura T, Hyodo I, et al. Phase I/II study of docetaxel and S-1 in patients with advanced gastric cancer. *Br J Cancer*. 2006;94(12):1803-1808.
- [5] Im CK, Jeung HC, Rha SY, et al. A phase II study of paclitaxel combined with infusional 5-fluorouracil and low-dose leucovorin for advanced gastric cancer. *Cancer Chemother Pharmacol*. 2008;61(2):315-321.
- [6] Lee KS, Lee HY, Park EK, Jang JS, Lee SJ. A phase II study of leucovorin, 5-FU and docetaxel combination chemotherapy in patients with inoperable or postoperative relapsed gastric cancer. *Cancer Res Treat*. 2008;40(1):11-15.
- [7] Park SR, Kim HK, Kim CG, et al. Phase I/II study of S-1 combined with weekly docetaxel in patients with metastatic gastric carcinoma. *Br J Cancer*. 2008;98(8):1305-1311.
- [8] Lee JJ, Kim SY, Chung HC, et al. A multi-center phase II study of S-1 plus paclitaxel as first-line therapy for patients with advanced or recurrent unresectable gastric cancer. *Cancer Chemother Pharmacol*. 2009;63(6):1083-1090.
- [9] Zang DY, Yang DH, Lee HW, et al. Phase I/II trial with docetaxel and S-1 for patients with advanced or recurrent gastric cancer with consideration to age. *Cancer Chemother Pharmacol*. 2009;63(3):509-516.
- [10] Kakeji Y, Oki E, Egashira A, et al. Phase II study of biweekly docetaxel and S-1 combination therapy for advanced or recurrent gastric cancer. *Oncology*. 2009;77(1):49-52.
- [11] Kunisaki C, Takahashi M, Makino H, et al. Phase II study of biweekly docetaxel and S-1 combination chemotherapy as first-line treatment for advanced gastric cancer. *Cancer Chemother*

Pharmacol. 2011;67(6):1363-1368.

[12] Shigeyasu K, Kagawa S, Uno F, et al. Multicenter phase II study of S-1 and docetaxel combination chemotherapy for advanced or recurrent gastric cancer patients with peritoneal dissemination. *Cancer Chemother Pharmacol.* 2013;71(4):937-943.

[13] Kosaka T, Akiyama H, Makino H, et al. Preoperative S-1 and docetaxel combination chemotherapy in patients with locally advanced gastric cancer. *Cancer Chemother Pharmacol.* 2014;73(2):281-285.

[14] Koizumi W, Kim YH, Fujii M, et al. Addition of docetaxel to S-1 without platinum prolongs survival of patients with advanced gastric cancer: a randomized study (START). *J Cancer Res Clin Oncol.* 2014;140(2):319-328.

[15] Oki E, Emi Y, Kusumoto T, et al. Phase II study of docetaxel and S-1 (DS) as neoadjuvant chemotherapy for clinical stage III resectable gastric cancer. *Ann Surg Oncol.* 2014;21(7):2340-2346.

[16] Jiang H, Qian J, Zhao P, et al. A phase II study of biweekly S-1 and paclitaxel (SPA) as first-line chemotherapy in patients with metastatic or advanced gastric cancer. *Cancer Chemother Pharmacol.* 2015;76(1):197-203.

[17] Kim YW, Kim MJ, Ryu KW, et al. A phase II study of perioperative S-1 combined with weekly docetaxel in patients with locally advanced gastric carcinoma: clinical outcomes and clinicopathological and pharmacogenetic predictors for survival. *Gastric Cancer.* 2016;19(2):586-596.

[18] Kosaka T, Akiyama H, Miyamoto H, et al. Outcomes of preoperative S-1 and docetaxel combination chemotherapy in patients with locally advanced gastric cancer. *Cancer Chemother Pharmacol.* 2019;83(6):1047-1055.

[19] Thuss-Patience PC, Kretschmar A, Repp M, et al. Docetaxel and continuous-infusion fluorouracil versus epirubicin, cisplatin, and fluorouracil for advanced gastric adenocarcinoma: a randomized phase II study. *J Clin Oncol.* 2005;23(3):494-501.

[20] Mochiki E, Ogata K, Ohno T, et al. Phase II multi-institutional prospective randomised trial comparing S-1+paclitaxel with S-1+cisplatin in patients with unresectable and/or recurrent advanced gastric cancer. *Br J Cancer.* 2012;107(1):31-36.

[21] Dai YH, Yu XJ, Xu HT, et al. Nab-paclitaxel plus S-1 versus oxaliplatin plus S-1 as first-line treatment in advanced gastric cancer: results of a multicenter, randomized, phase III trial (GAPSO study). *Ther Adv Med Oncol.* 2022;14:17588359221118020.

[22] Bian NN, Wang YH, Min GT. S-1 combined with paclitaxel may benefit advanced gastric cancer: Evidence from a systematic review and meta-analysis. *Int J Surg.* 2019;62:34-43.

[23] Japanese Gastric Cancer Association. Japanese gastric cancer treatment guidelines 2018 (5th edition). *Gastric Cancer.* 2021;24(1):1-21.

[24] Lin JL, Lin JX, Lin JP, et al. Safety and Efficacy of Camrelizumab in Combination With Nab-Paclitaxel Plus S-1 for the Treatment of Gastric Cancer With Serosal Invasion. *Front Immunol.* 2022;12:783243.

[25] Lin JX, Xu YC, Lin W, et al. Effectiveness and Safety of Apatinib Plus Chemotherapy as Neoadjuvant Treatment for Locally Advanced Gastric Cancer: A Nonrandomized Controlled Trial. *JAMA Netw Open.* 2021;4(7):e2116240.

[26] Zheng Y, Yang X, Yan C, et al. Effect of apatinib plus neoadjuvant chemotherapy followed by resection on pathologic response in patients with locally advanced gastric adenocarcinoma: A single-arm, open-label, phase II trial. *Eur J Cancer.* 2020;130:12-19.

[27] Tang Z, Wang Y, Yu Y, et al. Neoadjuvant apatinib combined with oxaliplatin and capecitabine in patients with locally advanced adenocarcinoma of stomach or gastroesophageal junction: a single-arm,

open-label, phase 2 trial. *BMC Med.* 2022;20(1):107.

[28] Jardim DL, Rodrigues CA, Novis YAS, Rocha VG, Hoff PM. Oxaliplatin-related thrombocytopenia. *Ann Oncol.* 2012;23(8):1937-1942.

[29] Al-Samkari H, Soff GA. Clinical challenges and promising therapies for chemotherapy-induced thrombocytopenia. *Expert Rev Hematol.* 2021;14(5):437-448.

3. Diffuse-type gastric cancer (DGC) was considered to be insensitive to chemotherapy and immune checkpoint inhibitors, and MPR of DGC was lower in DGC than intestinal-type gastric cancer either in CA-SAP or SAP group in this study. But it's inspiring that DGC derive higher benefit from neoadjuvant CA-SAP than SAP, with an MPR rate of 34.2% versus 12.5%. Is there a potential molecular mechanism to explain this efficacy?

Response: Thank you for your valuable comment. The unique biological characteristics and tumor microenvironment of diffuse-type gastric cancer make it less sensitive to chemotherapy and immunotherapy [1-3]. However, patients with diffuse-type tumors were more likely to benefit from CA-SAP in this trial. A feasible explanation was that the introduction of an anti-angiogenic agent (apatinib) altered the resistance profile of diffuse-type tumors. On one hand, anti-angiogenic therapy can improve the local hypoxia of diffuse-type tumors, thereby increasing sensitivity to chemotherapy and immunotherapy [4]. On the other hand, the immune-modulating properties of angiogenesis inhibitors may induce an immune-activated tumor microenvironment and enhance the efficacy of immunotherapy [5]. We have supplemented the above contents in the *Discussion* part of the revised manuscript.

Reference

[1] Jinawath N, Furukawa Y, Hasegawa S, et al. Comparison of gene-expression profiles between diffuse- and intestinal-type gastric cancers using a genome-wide cDNA microarray. *Oncogene.* 2004;23(40):6830-6844.

[2] Kim TS, da Silva E, Coit DG, Tang LH. Intratumoral Immune Response to Gastric Cancer Varies by Molecular and Histologic Subtype. *Am J Surg Pathol.* 2019;43(6):851-860.

[3] Pernet S, Terme M, Radosevic-Robin N, et al. Infiltrating and peripheral immune cell analysis in advanced gastric cancer according to the Lauren classification and its prognostic significance. *Gastric Cancer*. 2020;23(1):73-81.

[4] Mander KA, Finnie JW. Tumour angiogenesis, anti-angiogenic therapy and chemotherapeutic resistance. *Aust Vet J*. 2018;96(10):371-378.

[5] Shigeta K, Datta M, Hato T, et al. Dual Programmed Death Receptor-1 and Vascular Endothelial Growth Factor Receptor-2 Blockade Promotes Vascular Normalization and Enhances Antitumor Immune Responses in Hepatocellular Carcinoma. *Hepatology*. 2020;71(4):1247-1261.

4. In SAP cohort, CPS \geq 5 subgroup showed a higher MPR rate with 27.3% than 18.5% and 17.4% in patients with CPS \geq 1 and CPS $<$ 1 subgroups, respectively. how do you explain the association between neoadjuvant chemotherapy efficacy and PD-L1 expression?

Response: Thank you for your valuable comment. There is still no consensus regarding the association between PD-L1 expression and response to neoadjuvant chemotherapy. Some studies suggested that the CPS was not related to the efficacy of neoadjuvant chemotherapy [1-3]. However, Zurlo et al. found that a higher CPS (\geq 1%) was significantly associated with a higher MPR rate [4]. In this trial, although the MPR rate was higher in the CPS \geq 5% subgroup than in the CPS \geq 1% and CPS $<$ 1% subgroups, these differences did not reach statistical significance. Future studies are needed to confirm the relation between PD-L1 expression and response to neoadjuvant chemotherapy. We have supplemented the above contents in the *Discussion* part of the revised manuscript.

Reference

[1] Hashimoto T, Kurokawa Y, Takahashi T, et al. Predictive value of MLH1 and PD-L1 expression for prognosis and response to preoperative chemotherapy in gastric cancer. *Gastric Cancer*. 2019;22(4):785-792.

[2] Christina Svensson M, Lindén A, Nygaard J, et al. T cells, B cells, and PD-L1 expression in esophageal and gastric adenocarcinoma before and after neoadjuvant chemotherapy: relationship with histopathological response and survival. *Oncoimmunology*. 2021;10(1):1921443.

[3] Ribeiro HSC, Menezes JN, da Costa WL Jr, et al. PD-L1 expression in gastric and gastroesophageal junction cancer patients treated with perioperative chemotherapy. *J Surg Oncol*. 2022;126(1):150-160.

[4] Zurlo IV, Schino M, Strippoli A, et al. Predictive value of NLR, TILs (CD4+/CD8+) and PD-L1 expression for prognosis and response to preoperative chemotherapy in gastric cancer. *Cancer Immunol Immunother.* 2022;71(1):45-55.

5. the use of anti-VEGF drugs in peri-operative setting may be associated with safety concerns.

In this study, although the morbidity of anastomotic leakage was not statistically significant between two groups, CA-SAP group showed a trend toward more anastomotic leakage (8.2% versus 2%). I think this should be considered in large randomized clinical trials in the future.

Response: Thank you for your valuable suggestion. The anti-angiogenic effect of VEGF inhibitors may negatively affect anastomotic healing in gastrointestinal surgery [1]. In a randomized controlled trial involving 1063 patients with esophagogastric cancer, an increased incidence of anastomotic leakage was observed in the chemotherapy plus bevacizumab group (24%) than in the chemotherapy alone group (10%) [2]. However, apatinib plus neoadjuvant chemotherapy did not show a significant increase in the risk of anastomotic leakage in several prospective studies [3-5]. In this trial, although an increased incidence of anastomotic leakage was observed in the CA-SAP group (8.2%), this difference did not reach statistical significance. Thus, we recommended stopping apatinib treatment at least 14 days before surgery and correcting hypoalbuminemia/anemia during the perioperative course, which could minimize the risk of anastomotic leakage in patients treated with apatinib. We have supplemented the above contents in the *Discussion* part of the revised manuscript.

Reference

[1] Nakamura H, Yokoyama Y, Uehara K, et al. The effects of bevacizumab on intestinal anastomotic healing in rabbits. *Surg Today.* 2016;46(12):1456-1463.

[2] Cunningham D, Stenning SP, Smyth EC, et al. Peri-operative chemotherapy with or without bevacizumab in operable oesophagogastric adenocarcinoma (UK Medical Research Council ST03): primary analysis results of a multicentre, open-label, randomised phase 2-3 trial. *Lancet Oncol.*

2017;18(3):357-370.

[3] Zheng Y, Yang X, Yan C, et al. Effect of apatinib plus neoadjuvant chemotherapy followed by resection on pathologic response in patients with locally advanced gastric adenocarcinoma: A single-arm, open-label, phase II trial. *Eur J Cancer*. 2020;130:12-19.

[4] Xu Z, Hu C, Yu J, et al. Efficacy of Conversion Surgery Following Apatinib Plus Paclitaxel/S1 for Advanced Gastric Cancer With Unresectable Factors: A Multicenter, Single-Arm, Phase II Trial. *Front Pharmacol*. 2021;12:642511.

[5] Lin JX, Xu YC, Lin W, et al. Effectiveness and Safety of Apatinib Plus Chemotherapy as Neoadjuvant Treatment for Locally Advanced Gastric Cancer: A Nonrandomized Controlled Trial. *JAMA Netw Open*. 2021;4(7):e2116240.

Reviewers' comments:

Reviewer #1 (Remarks to the Author):

I am satisfied with the revised document.

Reviewer #2 (Remarks to the Author):

The authors have made some modifications to the review comments, which is worthy of confirmation, but some problems have been avoided. It seems unreasonable for a clinical trial to change the primary endpoint while the sample size remains unchanged. Please use reasonable statistical methods to explain, otherwise the scientific validity and rigor of the clinical trial will be questioned.

All patients with locally advanced proximal gastric cancer underwent total gastrectomy in this trial. In Table 2, it is shown that one patient in the CA-SAP group underwent proximal gastrectomy.

Please explain the immune-related adverse reactions in the SAP group.

Reviewer #3 (Remarks to the Author):

I don't have any issues with the statistical data analysis of the revised manuscript.

Reviewer #4 (Remarks to the Author):

The revised manuscript is much improved than the previous edition. I feel grateful for the authors's work and reply. however, I still have the following concerns.

Firstly, the retrospective factorial analysis was not that convincing to confirm the benefit of adding Apatinib in the Neoadjuvant treatment of gastric cancer, especially under the circumstances that KEYNOTE 585 failed to meet its primary endpoint DFS. Meanwhile, most of the references cited in this article were of low evidence. We can hardly change our clinical practice based on this present phase II clinical trial.

Secondly, translational research is very important in such phase II clinical trials, while the

authors did not do.

Therefore, I don't think this article can be published in Nature Communications.

REVIEWER COMMENTS

Reviewer #2 (Remarks to the Author):

The authors have made some modifications to the review comments, which is worthy of confirmation, but some problems have been avoided. It seems unreasonable for a clinical trial to change the primary endpoint while the sample size remains unchanged. Please use reasonable statistical methods to explain, otherwise the scientific validity and rigor of the clinical trial will be questioned.

Response: Thank you for your valuable comment. From the initial version (version 1.0) to the latest version (version 1.3) of the study protocol, the primary endpoint of this trial has remained as the major pathological response without changes. However, due to an error made by the trial registrant, the primary endpoint on clinicaltrials.gov was mistakenly registered as the pCR rate, which may have caused confusion. We sincerely apologize for any inconvenience caused. We have attached the study protocols for versions 1.0 and 1.3, as well as the summary of changes made in different versions.

All patients with locally advanced proximal gastric cancer underwent total gastrectomy in this trial. In Table 2, it is shown that one patient in the CA-SAP group underwent proximal gastrectomy.

Response: Thank you for your valuable comment. In this trial, total gastrectomy was scheduled 2 to 4 weeks after completion of the last cycle of neoadjuvant treatment for proximal gastric cancer patients. However, one patient in the CA-SAP group underwent palliative proximal gastrectomy due to acute bleeding. We have supplemented the above contents in the *Results* part of the revised manuscript.

Please explain the immune-related adverse reactions in the SAP group.

Response: Thank you for your valuable comment. Chemotherapy may have direct or indirect effects on immune cells, leading to immune-related adverse reactions [1-2]. Similar to the KEYNOTE-061, KEYNOTE-062, and ATTRACTION-4 trials [3-5], one immune-related adverse reaction was also observed in the SAP group, but its incidence was obviously lower than the CA-SAP group. We have supplemented the above contents in the *Discussion* part of the revised manuscript.

References

- [1] Chen G, Emens LA. Chemoimmunotherapy: reengineering tumor immunity. *Cancer Immunol Immunother.* 2013;62(2):203-216.
- [2] Dias Costa A, Väyrynen SA, Chawla A, et al. Neoadjuvant Chemotherapy Is Associated with Altered Immune Cell Infiltration and an Anti-Tumorigenic Microenvironment in Resected Pancreatic Cancer. *Clin Cancer Res.* 2022;28(23):5167-5179.
- [3] Kang YK, Chen LT, Ryu MH, et al. Nivolumab plus chemotherapy versus placebo plus chemotherapy in patients with HER2-negative, untreated, unresectable advanced or recurrent gastric or gastro-oesophageal junction cancer (ATTRACTION-4): a randomised, multicentre, double-blind, placebo-controlled, phase 3 trial. *Lancet Oncol.* 2022;23(2):234-247.
- [4] Shitara K, Özgüroğlu M, Bang YJ, et al. Pembrolizumab versus paclitaxel for previously treated, advanced gastric or gastro-oesophageal junction cancer (KEYNOTE-061): a randomised, open-label, controlled, phase 3 trial. *Lancet.* 2018;392(10142):123-133.
- [5] Shitara K, Van Cutsem E, Bang YJ, et al. Efficacy and Safety of Pembrolizumab or Pembrolizumab Plus Chemotherapy vs Chemotherapy Alone for Patients With First-line, Advanced Gastric Cancer: The KEYNOTE-062 Phase 3 Randomized Clinical Trial. *JAMA Oncol.* 2020;6(10):1571-1580.

Reviewer #4 (Remarks to the Author):

The revised manuscript is much improved than the previous edition. I feel grateful for the authors's work and reply. however, I still have the following concerns.

Firstly, the retrospective factorial analysis was not that convincing to confirm the benefit of adding Apatinib in the Neoadjuvant treatment of gastric cancer, especially under the circumstances that KEYNOTE 585 failed to meet its primary endpoint DFS. Meanwhile, most of the references cited in this article were of low evidence. We can hardly change our clinical practice based on this present phase II clinical trial.

Response: Thank you for your valuable comment. The phase II clinical trials are commonly conducted in small patient populations and aim to evaluate the safety and preliminary efficacy of new treatment strategies [1]. As you mentioned, the results of this phase II clinical trial may not be sufficient to change clinical practice. However, as the first randomized controlled trial on the advantages and safety of neoadjuvant immunotherapy and antiangiogenic therapy combined with chemotherapy in locally advanced gastric cancer, we believe that the results of this trial can provide important information for further research and serve as preliminary data for larger Phase III clinical trials.

References

[1] Simon R. Optimal two-stage designs for phase II clinical trials. *Control Clin Trials*. 1989;10(1):1-10.

Secondly, translational research is very important in such phase II clinical trials, while the authors did not do.

Response: Thank you for your valuable comment. Previous studies have summarized preclinical rationale on dual blockade combination with antiangiogenic agents and

immune checkpoint inhibitors [1]. Moreover, several clinical trials have reported significant associations between certain biomarkers and the efficacy of neoadjuvant immunotherapy and antiangiogenic therapy combined with chemotherapy [2-3]. We are also planning to develop a biomarker model to predict the efficacy of this combined treatment in a post hoc analysis. We have supplemented the above contents in the *Limitation* part of the revised manuscript.

References

[1] Saeed A, Park R, Sun W. The integration of immune checkpoint inhibitors with VEGF targeted agents in advanced gastric and gastroesophageal adenocarcinoma: a review on the rationale and results of early phase trials. *J Hematol Oncol.* 2021;14(1):13.

[2] Li S, Yu W, Xie F, et al. Neoadjuvant therapy with immune checkpoint blockade, antiangiogenesis, and chemotherapy for locally advanced gastric cancer. *Nat Commun.* 2023;14(1):8.

[3] Jing C, Wang J, Zhu M, et al. Camrelizumab combined with apatinib and S-1 as second-line treatment for patients with advanced gastric or gastroesophageal junction adenocarcinoma: a phase 2, single-arm, prospective study. *Cancer Immunol Immunother.* 2022;71(11):2597-2608.